# CXCL8, CCL2, and CMV Seropositivity as New Prognostic Factors for a Severe COVID-19 Course

**DOI:** 10.3390/ijms231911338

**Published:** 2022-09-26

**Authors:** Ewa Pius-Sadowska, Anna Niedźwiedź, Piotr Kulig, Bartłomiej Baumert, Anna Sobuś, Dorota Rogińska, Karolina Łuczkowska, Zofia Ulańczyk, Szymon Wnęk, Igor Karolak, Edyta Paczkowska, Katarzyna Kotfis, Miłosz Kawa, Iwona Stecewicz, Piotr Zawodny, Bogusław Machaliński

**Affiliations:** 1Department of General Pathology, Pomeranian Medical University, 70-111 Szczecin, Poland; 2Department of Anesthesiology, Intensive Therapy and Acute Intoxications, Pomeranian Medical University, 70-111 Szczecin, Poland

**Keywords:** COVID-19, SARS-CoV-2, chemokine, complement system, CMV status

## Abstract

The exact pathophysiology of severe COVID-19 is not entirely elucidated, but it has been established that hyperinflammatory responses and cytokine storms play important roles. The aim of this study was to examine CMV status, select chemokines, and complement components in COVID-19, and how concentrations of given molecules differ over time at both molecular and proteomic levels. A total of 210 COVID-19 patients (50 ICU and 160 non-ICU patients) and 80 healthy controls were enrolled in this study. Concentrations of select chemokines (CXCL8, CXCL10, CCL2, CCL3, CCR1) and complement factors (C2, C9, CFD, C4BPA, C5AR1, CR1) were examined at mRNA and protein levels with regard to a COVID-19 course (ICU vs. non-ICU group) and CMV status at different time intervals. We detected several significant differences in chemokines and complement profiles between ICU and non-ICU groups. Pro-inflammatory chemokines and the complement system appeared to greatly contribute to the pathogenesis and development of severe COVID-19. Higher concentrations of CXCL8 and CCL2 in the plasma, with reduced mRNA expression presumably through negative feedback mechanisms, as well as CMV-positive status, correlated with more severe courses of COVID-19. Therefore, CXCL8, CCL2, and CMV seropositivity should be considered as new prognostic factors for severe COVID-19 courses. However, more in-depth research is needed.

## 1. Introduction

The COVID-19 pandemic, which originated in Wuhan, China [1], and subsequently spread throughout the world, is believed to be one of the biggest challenges to healthcare systems worldwide. The vast majority of affected patients experience mild or moderate symptoms. The SARS-CoV-2 virus appeared to be highly mutagenic and contagious [2]. Thus, despite the low percentage of severe COVID-19 cases, the overall number of patients who required intensive medical care was significant. The initial symptoms of COVID-19 are, in most cases, similar to other viral infections that affect the respiratory tract; symptoms initially include fever, malaise, and dry cough. As the disease progresses, patients may show signs and symptoms of pneumonia, with or without hypoxemia. In critical conditions, acute respiratory distress syndrome may occur, with shock, coagulation disorders, encephalopathy, heart failure, and acute kidney injury. Moreover, some patients may present with gastrointestinal symptoms [3]. Loss of sense of smell and taste also appear to be critical symptoms of a COVID-19 diagnosis, particularly in the early stage of the disease [4]. The pathophysiology of severe COVID-19 is multifactorial. There are many known risk factors typically associated with a higher risk of developing severe disease. For instance, older age, hypertension, diabetes, obesity, chronic obstructive pulmonary disease, cancer, chemotherapy, and other chronic diseases, as well as immunodeficiencies associated with severe COVID-19 have been well-established [5]. Regardless of the patient’s medical histories and the aforementioned risk factors, individuals with severe infections tend to experience hyperinflammatory responses and cytokine storms [6].

A COVID-19 cytokine storm is predominantly associated with an innate immune system response [7]. Petrey and colleagues investigated cytokine release syndrome during the SARS-CoV-2 infection. Their study concluded that the cytokine profiles in COVID-19 cases are peculiar to innate responses (interleukin 6 (IL-6), interleukin 8 (IL-8/CXCL8), interleukin-1α (IL-1α), granulocyte colony-stimulating factor (G-CSF), growth-regulated oncogene-α (GROα/CXCL1), monocyte chemotactic protein-3 (MCP-3/CCL7), monocyte chemoattractant protein-1 (MCP-1/CCL2), and tumor necrosis factor-α (TNF-α). However, after further analysis, G-CSF, CXCL1, and IL-1α, considered separately, have become statistically nonsignificant, but collectively, they contribute to the overall effects. Moreover, the levels of several anti-inflammatory agents have also increased in comparison with the control group. However, an increase in anti-inflammatory cytokines failed to keep overall inflammation in check [8]. In particular, IL-6 seems to play an important role [9]. Its level was significantly higher upon admission in severe cases compared to mild cases. Moreover, severe COVID-19 patients who received high-flow oxygen inhalation and mechanical ventilation during hospitalization had significantly higher baseline IL-6 levels than those without the need for oxygen therapy support [10].

Abers et al. confirmed that CCL2 and interleukin-10 (IL-10) were associated with increased mortality in patients with COVID-19 [11]. Another study by Sacks et al. concluded that the highest expression levels (measured at mRNA level) of dipeptidyl peptidase 9 (*DPP9*) and *CCR2* were observed in severe forms of COVID-19 disease [12]. Similar observations were applied to the C-C motif chemokine ligand 3 (CCL3/MIP-1α) [13]. Given the above research, we focused on the following biomarkers—*CXCL8*, *CCL2*, and C-C motif chemokine receptor 1 (*CCR*1) expressions, as well as serum concentrations of CXCL8, C-X-C motif chemokine ligand 10 (CXCL10/IP-10), CCL2, and CCL3, as they appear to contribute to the pathogenesis of severe COVID-19.

Since there is ample evidence that hyperinflammation in COVID-19 is associated with the innate immune system, it could be hypothesized that complements also contribute to the pathology of COVID-19 clinical symptoms. It has been established that complement activation is frequent in COVID-19 patients and is presumably involved in the pathophysiology of clinical complications (e.g., lung and endothelial damage) in this subset of patients [14]. Magro et al. examined lung and skin tissues from patients with severe COVID-19 characterized by respiratory failure (n = 5) and purpuric skin rashes (n = 3). Their study revealed significant deposits of complements C5b-9, C4d, and mannose-binding lectin (MBL)-associated serine protease (MASP)-2 in the microvasculature. The purpuric skin lesions similarly showed a pauci-inflammatory thrombogenic vasculopathy, with depositions of complement C5b-9 and C4d in both grossly involved and normally-appearing skin. In addition, there was co-localization of COVID-19 spike glycoproteins with complements C4d and C5b-9 in the interalveolar septa and the cutaneous microvasculature activity in two cases examined. In conclusion, severe COVID-19 may define a type of catastrophic microvascular injury syndrome mediated by the activation of complement pathways and an associated procoagulant state [15].

The abovementioned evidence suggests that a severe SARS-CoV-2 infection is associated with hyperinflammation and a cytokine storm, which is mediated predominantly by the innate immune system. In particular, the role of the complement system seems to be interesting and is worth further exploration as its activation appears to mediate organ injury in severe COVID-19.

The role of cytokines in COVID-19 is well established. However, the exact courses of the immune processes remain terra incognita. Therefore, the aim of our study was to examine the roles of select chemokines as well as the complement system in COVID-19, and how concentrations of given molecules differ over time at both mRNA and proteomic levels.

Weber et al. conducted a study that investigated the relationship between disease severity and serological status of the herpes virus using statistical models. It was established that cytomegalovirus (CMV)-positive non-geriatric patients were more likely to develop severe COVID-19. Their study concluded that CMV seropositivity could be a potent biomarker in identifying younger individuals at a higher risk of developing severe COVID-19, especially in the absence of other comorbidities [16]. Therefore, we additionally investigated the status of CMV in our cohort and confronted it with chemokines and complement factors.

## 2. Results

### 2.1. Characteristics of the Study Group

The general characteristics of the study group were divided into SARS-CoV-2-negative controls and SARS-CoV-2-positive patients, including comorbidities and the severity of COVID-19 infection, as presented in Table 1.

The patients in the study group were (significantly) more often men with higher body mass indexes (BMIs). Patients with more severe courses of COVID-19 (ICU) were significantly older and had higher BMIs compared to non-ICU patients. We did not detect differences between men and women in terms of COVID-19 severity (*p* = 0.87). Interestingly, ICU patients were significantly less frequent cigarette smokers. Table 2 presents the clinical characteristics of the control group.

### 2.2. Expression of Select Chemokines at the mRNA Level

The expressions of chemokines *CXCL8*, *CCL2*, and *CCR1* were analyzed at the mRNA level at different time points. Figure 1 presents the relative expression levels of the listed chemokines among SARS-CoV-2-positive patients and SARS-CoV-2-negative controls.

The relative expression levels of *CXCL8* and *CCL2* were significantly higher in the control group than in the study group at all time points. However, from day 7 onwards, a virtually constant upward trend in the *CCL2* expression was apparent. The relative *CCR1* expression was significantly lower in the study group compared to the control (Figure 2).

*CXCL8* expression was significantly lower on days 1, 7, and 14, and the concurrent *CCL2* expression was significantly lower on days 1 and 7 in patients with more severe courses of COVID-19. *CCR1* expression was significantly lower only on day 7 in ICU patients.

In Figure 3, real-time quantitation of select chemokines is presented, depending on the CMV IgG status of SARS-CoV-2-positive patients and SARS-CoV-2-negative controls.

*CXCL8* expression was significantly lower on day 7 in CMV SARS-CoV-2-positive patients compared to CMV-negative patients.

### 2.3. Expressions of Selected Complement System Elements at the mRNA Level

As in the case of chemokines, the expressions of select elements of the complement system (*C4BPA*, *C5AR1*, *CFD*, and *CR1*) were analyzed at the mRNA level, and at various time points. In Figure 4, the relative gene expression levels are shown in patients with SARS-CoV-2 and the negative controls.

The relative expressions of *C4BPA, C5AR1*, and CFD were substantially higher in the study group at all time points compared to the control. *CR1* expression was significantly lower on day 7 and higher on day 28 in the study group compared to the control group.

Figure 5 presents a comparison of the relative expressions of complement system genes depending on the severity of the course of COVID-19.

*C4BPA* expression was higher at all time points in the ICU patients compared to non-ICU patients. *CR1* expression was substantially higher in the ICU patients on days 1 and 14 compared to non-ICU patients.

In Figure 6, the expressions of selected genes are presented, depending on the CMV IgG status of SARS-CoV-2-positive patients and SARS-CoV-2-negative controls.

*C5AR1* expression on day 14 was significantly higher in CMV-positive SARS-CoV-2-positive patients compared to CMV-negative patients.

### 2.4. Concentration of Select Chemokines at the Protein Level

In the next step of the research, select chemokines were assessed at the proteomic level. The concentrations of CXCL8, CXCL10, CCL2, and CCL3 in the plasma were analyzed at different time points. Figure 7 shows plasma chemokine levels in the whole population of SARS-CoV-2-positive patients and in negative controls.

In general, concentrations of CXCL10 and CCL3 were significantly higher at all time points in the study group compared to the control group. The plasma concentrations of CXCL8 on day 1 and CCL2 on day 7 were substantially higher in SARS-CoV-2-positive patients than in SARS-CoV-2-negative controls.

Figure 8 presents a comparison of plasma chemokine concentrations in patients depending on the severity of COVID-19.

ICU patients had significantly higher concentrations of all chemokines assessed in the plasma compared to non-ICU patients, up until day 7 of the observation. In the case of CXCL8, the concentration of chemokine in the plasma of ICU patients was higher at all time points compared to non-ICU patients.

Figure 9 shows the selected plasma chemokine concentrations depending on the CMV IgG status of SARS-CoV-2-positive patients and negative controls.

The mean concentrations of all assessed chemokines were higher on day 1 in CMV(+) SARS-CoV-2-positive patients compared to CMV(−) SARS-CoV-2-positive and SARS-CoV-2-negative patients, however statistically insignificant.

### 2.5. Concentration of Selected Complement System Elements at The Protein Level

The plasma concentrations of select elements of the complement system, analogous to chemokines, were assessed at various time points: C9, CFD, and C2. In Figure 10, the plasma concentrations of complement elements are shown in patients with SARS-CoV-2 and negative controls.

The baseline plasma concentrations of the tested complement elements were significantly higher in the population of SARS-CoV-2-positive patients compared to the control group. The plasma concentration of C9 on days 1, and 7, CFD at all time points, and C2 on day 1, were significantly higher in the study group compared to the control group. Concentrations of C9 on day 28 and C2 on day 14 were significantly lower in SARS-CoV-2-positive patients compared to SARS-CoV-2-negative patients.

Figure 11 compares the plasma levels of the complement system in patients according to the severity of COVID-19.

Plasma concentrations of C9 were higher in ICU patients on days 1 and 14 compared to non-ICU patients. Concentrations of CFD were substantially lower in ICU patients on days 7 and 14 compared to non-ICU patients.

Figure 12 shows the plasma concentrations of the selected complement elements versus the CMV IgG status of SARS-CoV-2-positive patients and negative controls.

We observed very high concentrations of C9 on days 1 and 7 in the CMV(+) and CMV(−) SARS-CoV-2-positive group compared to the control. However, the differences between the CMV(+) and CMV(−) SARS-CoV-2-positive patients were statistically insignificant. On days 14 and 28, a sharp decrease in C9 concentration can be observed with significant differences between the CMV(+) and CMV(−) groups.

### 2.6. CMV Status

Subsequently, we assessed CMV status within our cohort. We observed several statistically significant differences in the relative expressions and concentrations of both chemokines and complement components at molecular and protein levels. Table 3 and Table 4 show the differences between day 1 and the following time points in chemokine and complement mRNA expressions and protein concentrations in the patient groups, CMV(+) and CMV(−) respectively.

In the control group, 83% were CMV(+) and 17% CMV(−). In the ICU group, 98% were CMV(+) and 2% CMV(−). In the non-ICU group, 85% were CMV(+) and 15% CMV(−). We also checked whether CMV status was associated with COVID-19 severity. Nearly every patient in the ICU group was CMV-positive, and there were significant differences between the ICU and non-ICU groups with regard to the CMV serological status (Table 5).

Positive CMV statuses of ICU patients correlated with more severe courses of COVID-19.

## 3. Discussion

The role of cytokines in severe COVID-19 is very well established, and individuals with severe disease, despite certain similarities with mild or moderate courses, exhibit different cytokine profiles. We intend to further explore the pathogenesis of COVID-19-related inflammation, so our study focused on the cytokine subtype–chemokines and complement systems at both mRNA and proteomic levels. We demonstrated that ICU and non-ICU patients exhibited distinct and unique chemokine and complement profiles. CXCL8 and CCL2 were of particular importance (with regard to the disease course); their significance is discussed below.

Chemokines are a subtype of cytokines that act as chemoattractants, i.e., mediate the migration of leukocytes to the site of inflammation [17,18] Nowadays, approximately 50 endogenous chemokine ligands and 20 G-protein-coupled receptors have been described [19]. In our study, we examined the chemokine profiles at both mRNA and proteomic levels in ICU and non-ICU COVID-19 patients at different time intervals. From the pathophysiological point of view, the analysis of CXCL8 and CCL2 chemokines provided particularly important data, as the analysis was carried out at both the mRNA and protein levels. The remaining chemokines were only assessed at the molecular or proteomic level, which made it difficult to draw conclusions.

Our study confirmed that the relative expressions of *CXCL8* and *CCL2* in SARS-CoV-2-positive patients were significantly lower than in the control group at most time points. On the other hand, the plasma concentrations of CXCL8 on day 1 and CCL2 on day 7 were significantly higher in the SARS-CoV-2-positive group than in the control group. Lower baseline *CXCL8* and *CCL2* expressions correlated with more severe courses of COVID. Increased CXCL8 plasma concentrations on all days and CCL2 on days 1 and 7 also correlated with more severe courses of COVID. These seemingly contradictory processes at the molecular and protein levels can be explained in some way. The baseline level of cytokines was assessed in patients with full-blown COVID-19 infection. The chemokines we study were produced in humans by macrophages, T lymphocytes, neutrophils, fibroblasts, osteoblasts, endothelial cells, and epithelial cells. During acute infectious symptoms, the previously produced chemokines were released into the circulation. The increase in plasma chemokine concentration, through a negative feedback mechanism, most likely suppressed mRNA expression. Therefore, only a gradual decrease in the concentration of chemokines in the following days of observation resulted in a parallel increase in mRNA expression. This hypothesis seems to be supported by Abers et al., who found a similar association to ours, noting that the discordance between the IFN-α2a protein and IFNA2 transcript blood level suggests that type I IFNs during COVID-19 may be primarily produced by tissue-resident cells. Moreover, the results of this study revealed an association between disease severity and CCL2, which was also confirmed by our results [11]. Additionally, Arunachalam et al. showed that IFN-stimulated genes showed transient expressions during COVID-19 infection in a time-dependent manner [20].

CXCL8 is a pro-inflammatory chemokine and is considered one of the major chemoattractants for neutrophils [21]. High serum levels of CXCL8 are commonly found in conditions where inflammation is involved. Moreover, it might be associated with poorer clinical outcomes in a large variety of medical conditions. For instance, Shetelig and colleagues conducted a study that investigated the role of CXCL8 in patients with acute ST-segment elevation myocardial infarction (STEMI). They obtained serum samples from 258 patients with STEMI at different time intervals. Their study revealed that high levels of circulating CXCL8 were associated with large infarct sizes, impaired recovery of the left ventricle function, and adverse clinical outcomes in patients with STEMI [22]. In another study, the predictive value of the CXCL8 serum level in burn patients was established. High CXCL8 levels were able to predict sepsis (*p* < 0.002). In the group with a high CXCL8 level, elevated CXCL8 was associated with increased inflammatory and acute phase responses compared to the low CXCL8 group (*p* < 0.05). High levels of CXCL8 correlated with increased multiorgan failure, sepsis, and mortality. These data suggest that CXCL8 serum levels may be valid biomarkers for monitoring sepsis, infection, and mortality in burn patients [23]. High levels of IL-6 and CXCL8 are also associated with poorer clinical outcomes in sickle cell anemia [24].

CXCL8 was also investigated in COVID-19. Higher plasma concentrations of CXCL8 were associated with greater mortality [25]. In another study, ICU patients had significantly greater expressions of pro-inflammatory chemokine transcripts *CXCL1* (*p* = 0.03) and *CXCL8* (*p* = 0.04) compared to outpatients. Our research showed the opposite relationship with *CXCL8* expression. Perhaps the research was conducted at a different phase of the COVID-19 infection (hence, the discrepancies). The expression of the *CXCL5* gene was also enhanced in ICU patients, but the difference did not reach significance (*p* = 0.10) [26]. Blot and colleagues investigated the cytokine profiles in non-COVID-19 and COVID-19 patients with severe pneumonia. Interestingly, COVID-19 patients had lower levels of most classic inflammatory cytokines, including CXCL8. This also contradicts our research. However, SARS-CoV-2-positive individuals had higher plasma concentrations of CXCL10, CCL5, and GM-CSF, which was also associated with a longer duration of mechanical ventilation [27]. Due to the fact that there is no consensus in the literature and many authors describe contradictory results, there is a need for further exploration of this area.

Stikker et al. investigated the 3p21.31 locus and subsequent downstream signaling with regard to the COVID-19 severity. They demonstrated that several 3p21.31 variants, previously identified by applying genome-wide association studies, were associated with increased *CCR1*, *CCR2*, *CCR3*, and *CCR5* expressions. *CCR1*, *CCR2*, and *CCR5* upregulations could enhance lung infiltration by monocytes and macrophages upon viral infection and mediate hyperinflammation and organ damage in the aftermath [28]. However, we did not observe such an association.

Contrary to CXCL8, IL-10 exerts anti-inflammatory properties. Therefore, it plays a key role in infection, limiting the immune response to pathogens and, thus, preventing damage to the host [29]. IL-10 both modulates the local cytokine microenvironment and limits antigen presentation, thus preventing the efficient development of T-cell responses [30]. This could be of paramount importance as exacerbated cytotoxicity was linked with fatal COVID-19 outcomes [31].

It was established that serum concentrations of IL-10 were higher in severe COVID-19 patients compared to the recovered and control groups [32]. Moreover, high serum levels of cytokines, such as IL-6, CCL2, CXCL8, CXCL10, IL-2, and IL-10 were associated with COVID-19-related encephalopathy and showed an enhanced systemic inflammatory response [33]. Due to the anti-inflammatory properties of IL-10, it can be hypothesized that the relationship between its elevated level in the severe manifestation of the disease may be a protective mechanism aimed at counteracting the cytokine storm and the hyperinflammatory response in ICU patients. However, further research is necessary to address this issue.

As mentioned above, the innate immune system, with a particular emphasis on cytokines and chemokines, appears to be a key player in mediating organ damage in severe COVID-19. Our results were not only coherent with current knowledge, but also provided more specific insights on the pathophysiology of severe COVID-19. According to the study conducted by Lucas and team, the cytokine profile partially overlaps in patients with moderate and severe disease. It was established that these two subsets of patients share the following inflammatory profile defined by IL-1α, IL-1β, IL-17A, IL-12 p70, and interferon-α (IFN-α). Nevertheless, in the severe disease there was observed an additional inflammatory cluster, characterized by thrombopoietin (TPO), IL-33, IL-16, IL-21, IL-23, IFN-λ, eotaxin and eotaxin 3, which distinguishes these patients from those with mild or moderate disease. Most of the cytokines linked to cytokine release syndrome (CRS), such as IL-1α, IL-1β, IL-6, IL-10, IL-18, and TNF-α, showed an increased positive association with the severity of the disease. Further analysis revealed that IFN-α levels were sustained at higher levels in severe disease patients while they declined in the subgroup with moderate patients. Plasma IFN-λ levels increased during the first week of symptom onset in ICU patients and remained elevated in later phases. Additionally, inflammasome-induced cytokines, such as IL-1β and IL-18 were also elevated in severe disease compared to moderate disease patients. IL-6, which is linked to CRS, was elevated in patients with severe disease [34]. In another study, an impaired type I IFN response was detected in severe and critical COVID-19 patients, accompanied by high blood viral load and an excessive nuclear factor kappa-light-chain-enhancer of activated B cells (NF-κB)–driven inflammatory response characterized by increased TNF-α and IL-6. The authors hypothesize that their results suggest that SARS-CoV-2 has developed efficient mechanisms to shut down host IFN production [35]. Dean et al. also described impaired type I IFN response in severe COVID-19. In their study, a comparison between severe and asymptomatic patients showed that severe patients had significantly decreased expression of genes from pathways associated with type I IFN responses, which reinforces the concept that interferon-stimulated gene (ISG) pathways are critical for a protective immune response to SARS-Cov-2 [36].

The complement system seems to play a pivotal role in the pathogenesis of severe SARS-CoV-2 infection and at least partially mediate organ damage [14,15]. We also detected significant differences between ICU and non-ICU groups in the mRNA expressions and plasma concentrations of the selected complement factors. In our study, similar to chemokines, the most interesting observation concerned CFD, which was assessed at both mRNA and proteomic levels. CFD expression was significantly higher in SARS-CoV-2-positive patients on all tested days than in the control group. These results correlated with the plasma concentrations of CFD, which were substantially higher in SARS-CoV-2-positive patients than in SARS-CoV-2-negative controls at all time points. There was no correlation between the ICU and non-ICU groups in terms of CFD expression. However, plasma CFD concentrations were significantly lower in the ICU group on days 7 and 14 than in the non-ICU population.

Georg et al. established that the generation of CD16^+^ T cells and subsequent complement activations were associated with the severe manifestation of COVID-19 and contributed to its pathogenesis. In addition, the authors hypothesize that this functionally links the innate and adaptive immune system with endothelial injury, which could constitute an important molecular axis explaining the vast spectrum of organ damage observed in COVID-19 [31]. Carvelli and colleagues demonstrated the role of the C5a-C5AR1 axis in the pathophysiology of acute respiratory distress syndrome (ARDS) in severe COVID-19 [37]. In our study, the expressions of *C5AR1* were significantly higher at all time points in the SARS-CoV-2 positive group compared to the control group. However, we did not observe any significant difference in *C5AR1* expression between the ICU and non-ICU groups. In addition, complement activation mediates, at least partially, platelet aggregation and immunothrombosis in severe COVID-19 [38].

We examined the serological status of CMV in our cohort. The study revealed differences between CMV-positive and CMV-negative patients in both chemokine and complement factor profiles at mRNA and protein levels. However, these differences were not statistically significant in the vast majority of cases, probably due to a small CMV-negative cohort. It is worth mentioning that CMV(+) individuals accounted for 98% of the ICU patients, 85% of the non-ICU patients, and 83% of the control group. Simultaneously, the ICU and non-ICU groups significantly differed in terms of CMV serological status (*p* = 0.01). We have unequivocally shown that a positive CMV status correlates with a more severe course of COVID-19. CMV infection affects the immune system. For instance, CMV seropositive individuals have been shown to exhibit peculiar CD4^+^ T cell subsets compared to CMV-negative patients [39]. Moreover, a latent CMV infection accelerates aging-related changes in the immune system in transplant recipients, such as impaired T cell proliferation and signaling, as well as impaired vaccine responses. This phenomenon implies that age-related conditions may affect this subset of patients at a younger age than expected [40]. Since the interaction between CMV infection and the immune system is well established, it may be hypothesized that it also affects SARS-CoV-2 infection and may increase its severity.

Currently, very little is known about the influence of CMV status on the course of COVID-19. Nonetheless, it was the subject of scientific research. Weber and colleagues investigated CMV status in patients who experienced mild, moderate, or severe critical disease. Their study revealed that CMV seropositivity could be a potential novel risk factor for severe COVID-19 in the non-elderly individuals in the studied cohorts [16]. As their results were novel, we decided to further explore this area. In our study, SARS-CoV-2-positive patients were mostly middle-aged, and our observations confirm the above relationship. Moreover, we believe that as this phenomenon is not yet well established in the literature, our results contribute to the field and shed more light on the pathophysiology of severe COVID-19.

In a study conducted by Alanio and colleagues, CMV seropositivity was associated with an increased risk of SARS-CoV-2 infection and hospitalization [41]. Again, our data correspond with the literature. It is no coincidence that nearly all ICU patients were CMV-positive (98%). Nevertheless, CMV status and its impact on the course of SARS-CoV-2 infection should be further investigated.

Although the pathogenesis of severe COVID-19 is not entirely elucidated, our results as well as the literature data reveal that innate immunity plays an important role in the pathogenesis of severe COVID-19. We demonstrated that concentrations of both pro-inflammatory and anti-inflammatory cytokines differ between ICU and non-ICU patients at both mRNA and protein levels. However, mRNA expression does not always correspond with proteomics. Perhaps this is the effect of the aforementioned primary release of chemokines into the circulation followed by negative feedback. On the other hand, the complement elements showed a direct correlation between mRNA and protein.

Immune system dysregulation, which may also be enhanced by a latent CMV infection, is accompanied by activation of the complement system. Complement’s role in inflammation is thoroughly described in the literature. It acts as a molecular clip that binds both innate and adaptive immunity [42]. The above interplay could be a potential therapeutic target. For instance, therapeutic anti-C5AR1 monoclonal antibodies have prevented C5a-mediated human myeloid cell recruitment and activation and inhibited acute lung injury in a mouse model [37]. In addition, a clinical trial (phase 2 study) with zilucoplan (complement C5 inhibitor) is ongoing to assess the efficacy and safety of zilucoplan in improving oxygenation and short- and long-term outcomes in COVID-19 patients with acute hypoxic respiratory failure [43].

In conclusion, we demonstrated immune system dysregulation in severe COVID-19, as exemplified by the overactivation of specific chemokines and complement components at both the molecular and protein levels. The main conclusions of our work are as follows:-Lower mRNA expression and higher concentrations of CXCL8 and CCL2 in the plasma correlated with more severe courses of COVID-19.-CMV-positive status correlated with a more severe COVID-19 course.-CXCL8, CCL2, and CMV seropositivity should be considered as new prognostic factors for severe COVID-19 courses.

Pro-inflammatory chemokines and the complement system appear to greatly contribute to the pathogenesis and development of severe COVID-19. However, further in-depth studies are needed to fully address this issue.

## 4. Materials and Methods

### 4.1. Study Group

This retrospective cohort study included 210 patients diagnosed with COVID-19 at the Department of Infectious, Tropical Diseases and Acquired Immunodeficiency, Pomeranian Medical University in Szczecin, Poland. SARS-CoV-2 infection was confirmed from nasopharyngeal swabs with a real-time polymerase chain reaction (RT–PCR) technique. The control group consisted of 80 healthy individuals with negative RT–PCR results for SARS-CoV-2 obtained from nasopharyngeal swabs and negative ELISA results for SARS-CoV-2-specific IgG, IgM, and IgA antibodies; participants were recruited from the hospital staff. Both groups enrolled in the study underwent blood collection to determine the expressions of predefined genes and concentrations of select chemokines and complement components; they were asked to complete detailed questionnaires regarding their general health. The study was approved by the Ethics Committee of the Pomeranian Medical University in Szczecin (KB-0012/83/2020) and conducted in accordance with the Declaration of Helsinki. An informed consent form was signed by each participant before study enrolment.

### 4.2. General Health Questionnaire

All patients were interviewed and examined to collect information on the presence of signs and symptoms, such as fever, dyspnea, cough, cold, sore throat, fatigue, chest pain, smell/taste abnormalities, headache and body aches, or diarrhea, and the severity and duration of the abovementioned symptoms. Data regarding the laboratory findings, the need for oxygen or respiratory-assisted therapy, the presence of pneumonia on chest computed tomography (CT), the need for hemodialysis, or the patient’s death, were collected from electronic medical records. Demographics, family history of various diseases, and other general health risk factors present at the time of the enrolment, such as hypertension, hyperlipidemia, smoking, diabetes, cardiovascular, liver, respiratory and rheumatic diseases, and previous cerebrovascular events, were recorded. The severity of COVID-19 was retrospectively assessed. All enrolled COVID-19-positive patients were divided into two groups based on disease severity. Group 1 (intensive care unit patients—ICU patients) included patients who required ICU admission due to respiratory failure, hospitalization longer than 14 days directly related to COVID-19, or where the disease was fatal. Meeting each criterion, individually or in combination, determined the inclusion of patients in the ICU group. Group 2 (non-ICU patients) included asymptomatic or mildly symptomatic patients (who had oxygen saturation of at least 95% and did not require hospitalization due to COVID-19) and symptomatic patients with oxygen saturation lower than 95% (who required hospital admission up to fourteen days).

### 4.3. Material

#### 4.3.1. Plasma Collection

Peripheral blood samples were collected upon admission to the hospital (day 1) and during hospitalization/isolation on days 7, 14, and 28 after COVID-19 diagnosis. Peripheral blood samples (~7.5 mL) collected in EDTA tubes were centrifuged (2000 rpm, 10 min). Then, the plasma was collected in a new tube and centrifuged again under the same conditions. The plasma samples were stored at −80 °C. Plasma-free blood was lysed with Lysing Solution (BD Biosciences, San Jose, CA, USA) for 15 min at room temperature to obtain blood mononuclear cells.

#### 4.3.2. RNA Isolation

##### Viral RNA Isolation

Viral RNA isolation was performed using the MagMAX Viral/Pathogen II Nucleic Acid Isolation Kit (Thermo Fisher Scientific, ON, CA) according to the manufacturer’s protocol. A total of 200 μL of each sample was added to the designated sample well and mixed with 5 μL of proteinase K, 5 μL of MS2 phage control, 265 μL of binding buffer, and 10 μL of magnetic beads. A total of 200 μL of nuclease-free water was also pipetted into the negative control well in the sample plate. Furthermore, 3 processing plates (KingFisher 96 Deep-Well Plate, Thermo Fisher Scientific, ON, CA) with Wash 1 Solution (500 µL per well), Wash 2 Solution—80% ethanol (1000 µL per well), and Elution Solution (50 µL per well) were prepared. MagMAX Viral/Pathogen nucleic acid isolation was processed using an automated KingFisher Flex instrument (Thermo Fisher Scientific, ON, CA).

#### Blood Mononuclear Cell RNA Isolation

Total RNA was isolated from blood mononuclear cells using the commercial mirVana miRNA Isolation Kit (Thermo Fisher Scientific, Waltham, MA, USA). The isolation was performed according to the manufacturer’s protocol. The final concentration and quality of total RNA isolated from the cells were determined by an Epoch spectrophotometer (BioTek, Winooski, VT, USA).

#### 4.3.3. qRT-PCR

##### qRT–PCR Assays for Detecting SARS-CoV-2 RNA

RT–qPCR assays for the detection of SARS-CoV-2 RNA were performed using a QuantStudio 5 PCR instrument an *CXCL8* expression was significantly d a TaqPath COVID 19 CE IVD RT PCR Kit (Thermo Fisher Scientific, Markham, ON, CA) according to the manufacturer’s protocols. The one-step RT–qPCR contained 5 μL of RNA template/TaqPath COVID 19 Control, 6.25 μL of 4 × TaqPath 1 Step Multiplex Master Mix (Thermo Fisher Scientific, Markham, ON, CA), 1.25 µL of COVID-19 Real-Time PCR Assay Multiplex, and 7.5 µL of nuclease-free water in a total volume of 20 μL. The one-step RT–qPCR program included the RT reaction at 53 °C for 10 min, enzyme activation at 95 °C for 2 min, 40 cycles of PCR amplification at 95 °C for 3 s, and 60 °C for 30 s. After RT–PCR was completed, the results were analyzed using Applied Biosystems COVID-19 Interpretive Software (Thermo Fisher Scientific, Markham, ON, CA). Those tested were considered positive when at least 2 out of 3 analyzed SARS-CoV-2 genes (ORF1ab, N, S) had Ct values ≤ 37.

##### qRT–PCR Assays for Evaluating Complement and Chemokine mRNA Expressions

Primers for selected genes (*CXCL8*, *CCL2*, *CCR1*), complement component 4 binding protein alpha (*C4BPA*), complement component 5a receptor 1 (*C5AR1*), complement factor D (*CFD*), and complement receptor type 1 (*CR1*) were designed by BLAST PRIMER and purchased from the Laboratory of DNA Sequencing and Oligonucleotide Synthesis, Institute of Biochemistry and Biophysics, Polish Academy of Sciences, Warsaw, Poland. The following primer sequences were used: *hCXCL8*, f: 5′-TTCAGAGACAGCAGCAGAGCACA-3′, r: 5′-AGCACTCCTTGGCAAAACTG-3′; *hCCL2*, f: 5′- GATCTCAGTGCAGAGGCTCG-3′, r: 5′-TTTGCTTGTCCAGGTGGTCC -3′; *hCCR1*, f: 5′- AGAAGCCGGGATGGAAACTC-3′, r: 5′-TTCCAACCAGGCCAATGACA-3′; *hC4BPA*, f: 5′-AGGGACTCTTTGGTGGAGCA-3′, r: 5′-CTGCTGCTTCGCTGATGTTT-3′; *hC5AR1*, f: 5′-AGCCCAGGAGACCAGAACAT-3′, r: 5′-CACCAGGAAGACGACTGCAA-3′; *hCFD*, f: 5′- GATGTGCGCGGAGAGCAAT-3′, r: 5′-CTGTCGATCCAGGCCGCATA-3′; *hCR1*, f: 5′-TCTGCTGTCTTGGGTGCATT-3′, r: 5′-TTCGTGATGATTCTGCCCCC-3′; *hBMG*, f: 5′-AATGCGGCATCTTCAAACCT-3′, r: 5′-TGACTTTGTCACAGCCCAAGA-3′. The qRT-PCR program consisted of 4 steps: 10-min initial denaturation at 95 °C, denaturation at 95 °C for 15 s, annealing at 60 °C for 5 s, and extension at temperature (depending on the selected primer for 10 s). The relative gene expression was quantified using the comparative Ct method 2ΔCt. *Beta*-2-Microglobulin (BMG) was set as the reference gene. All products were characterized by high specificity and were checked by determining the melting points (0.1 °C/s transition rate). Reverse transcription was conducted via the First Strand cDNA Synthesis Kit (Thermo Fisher Scientific, Waltham, MA, USA); the qRT-PCR reaction mixture (10 μL) contained 5 μL of SYBR Green PCR Master Mix (Bio-Rad, Hercules, CA, USA), 1 μL cDNA template, 1.2 μL specific primers (0.6 μL reverse primer and 0.6 μL forward primer), and 2.8 μL Nuclease-Free Water. The gene expression studies were performed on the Bio-Rad CFX96 Real-Time PCR Detection System (Bio-Rad Inc., Hercules, CA, USA).

#### 4.3.4. Luminex Assay

The concentrations of chemokines, such as CXCL8, CXCL10, CCL2, and CCL3, and the complement components C9, CFD, and C2, were measured in plasma by multiplex fluorescent bead-based immunoassays (Luminex Corporation, Austin, TX, USA) using commercial R&D Systems Luminex Human Discovery Assay (R&D Systems, Minneapolis, MN, USA). A total of 50 µL of blanks, standards, and samples were added to the plate along with the Microparticle Cocktail and incubated in the dark for 2 h at room temperature on a horizontal orbital microplate shaker set at 800 rpm. After this step, the wells were washed three times with 100 µL of wash buffer using a hand-magnet.

A biotin-antibody cocktail (50 µL) was added to the plate and incubated with agitation at room temperature for 60 min in the dark. After washing, 50 µL of streptavidin–PE was added to each well and incubated in the dark for 30 min on a plate shaker. Finally, after washing, the microspheres in each well were resuspended in 100 µL of wash buffer and shaken for 2 min at room temperature. The plate was read and analyzed on the Luminex 200 analyzer, and analyte concentrations were determined from seven different standard curves showing median fluorescence intensity vs. protein concentration.

#### 4.3.5. Enzyme-Linked Immunosorbent Assay (ELISA)—Serological Assays for Specific Anti-SARS-CoV-2 IgM, IgG, IgA, and Anti-CMV IgG Antibodies Detection

All of the plasma samples were tested for the presence of anti-SARS-CoV-2 immunoglobulins (IgM, IgG, IgA) and anti-CMV IgG antibodies using a commercially available enzyme-linked immunosorbent assay (EUROIMMUN AG, Lubeck, Germany). First, the reagent wells on the microplate strips that had previously been coated with inactivated CMV antigen and recombinant structural spike protein of SARS-CoV-2 were filled with the diluted patient plasma sample and allowed to incubate. Specific anti-CMV IgG and anti-SARS-CoV-2 IgM, IgG, and IgA antibodies, attached to the coated wells, were identified with peroxidase-labeled anti-human IgM/IgG/IgA antibodies. The staining intensity generated after hydrolysis of the peroxidase substrate was measured using a Varioskan LUX multimode microplate reader (TSA, Thermo Fisher Scientific, Waltham, MA, USA) at a wavelength of 450 nm. The results were interpreted according to the manufacturer’s instructions: samples with concentrations higher than 22 RU/mL were considered positive for CMV IgG, while samples below the cut-off value were considered negative. The results for SARS-CoV-2 specific antibodies were evaluated semi-quantitatively by calculating the ratio (R) of the extinction of the control or patient sample over the extinction of the calibrator. Following the manufacturer’s instruction, the level of positivity was calculated as the R between the absorbance values of samples and the calibrator, at a wavelength of 450 nm (R < 0.8, negative; 0.8 < R < 1.1, weakly positive; R ≥ 1.1, strongly positive).

### 4.4. Statistical Analysis

The Mann–Whitney test was used in our analyses to compare the quantitative parameters between the groups. Fisher’s exact test was implemented to assess the differences between the categorical variables. Friedman ANOVA and the subsequent post hoc Wilcoxon signed-rank test with Bonferroni correction were used to assess differences between the given time points within the group. A *p* value of < 0.05 was considered statistically significant. All calculations were performed in RStudio version 1.2.1335.

## 5. Study Limitations

Overall, our study provided some interesting results, but it included some drawbacks. Although the study and control groups were numerous, they differed significantly in terms of sex and BMI. This was the result of selecting a control group that consisted of hospital employees (mostly women). Another disadvantage was the evaluation of different chemokines and complement elements at the molecular and protein levels, which made it difficult to draw conclusions. This limitation was due to the commercial availability of laboratory tests on strictly defined sets of chemokines and complement elements only.

## Figures and Tables

**Figure 1 ijms-23-11338-f001:**
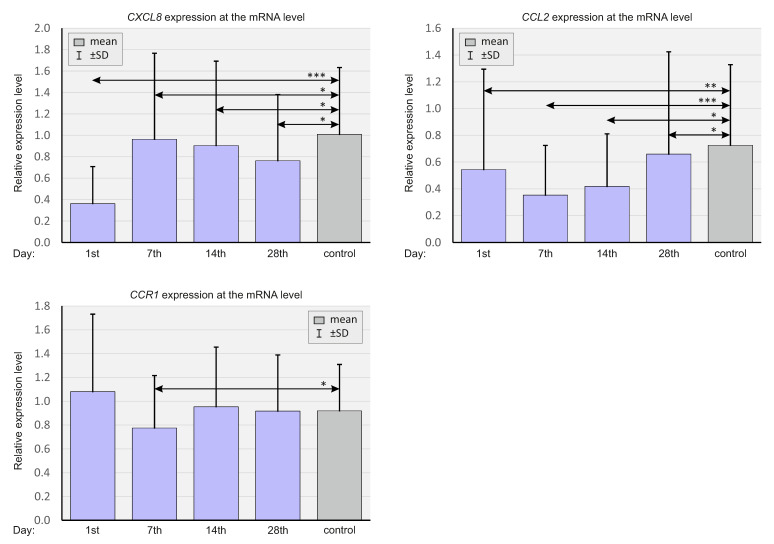
Bar graphs showing real-time quantitation of selected genes *CXCL8, CCL2,* and *CCR1* in the whole population of SARS-CoV-2-positive patients and SARS-CoV-2-negative controls. *p* < 0.05—*, *p* < 0.001—**, *p* < 0.0001—***.

**Figure 2 ijms-23-11338-f002:**
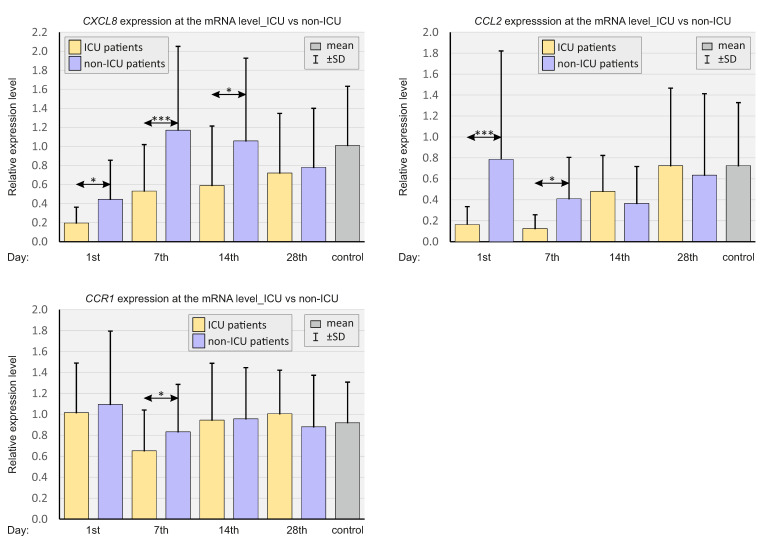
Bar graphs showing real-time quantitation of the selected genes *CXCL8, CCL2,* and *CCR1* in patients depending on the severity of the course of COVID-19 and in SARS-CoV-2-negative controls. *p* < 0.05—*, *p* < 0.0001—***.

**Figure 3 ijms-23-11338-f003:**
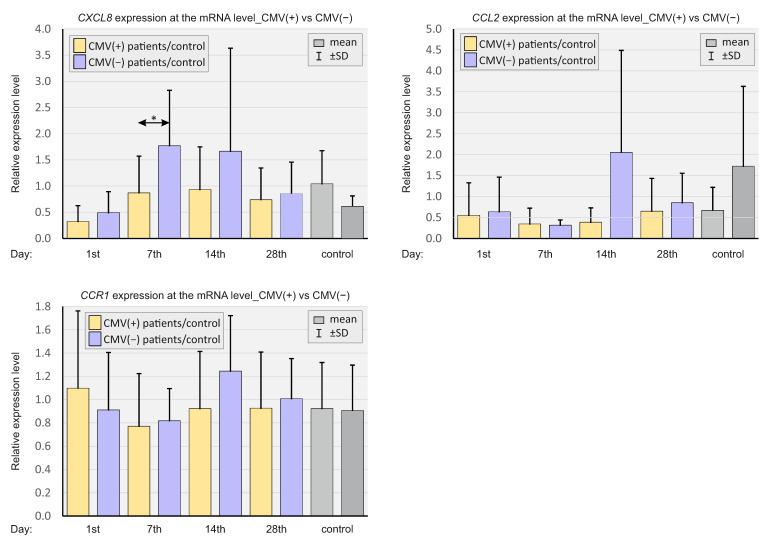
Bar graphs showing real-time quantitation of selected genes *CXCL8*, *CCL2*, and *CCR1*, depending on the CMV status in SARS-CoV-2-positive patients and SARS-CoV-2-negative controls. *p* < 0.05—*.

**Figure 4 ijms-23-11338-f004:**
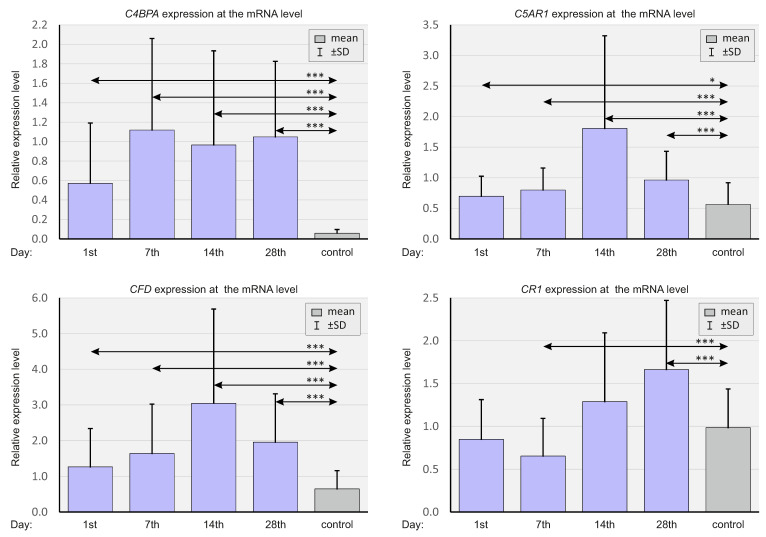
Bar graphs showing real-time quantitation of selected genes *C4BPA*, *C5AR1*, *CFD*, and *CR1* in the whole population of SARS-CoV-2-positive patients and SARS-CoV-2-negative controls. *p* < 0.05—*, *p* < 0.0001—***.

**Figure 5 ijms-23-11338-f005:**
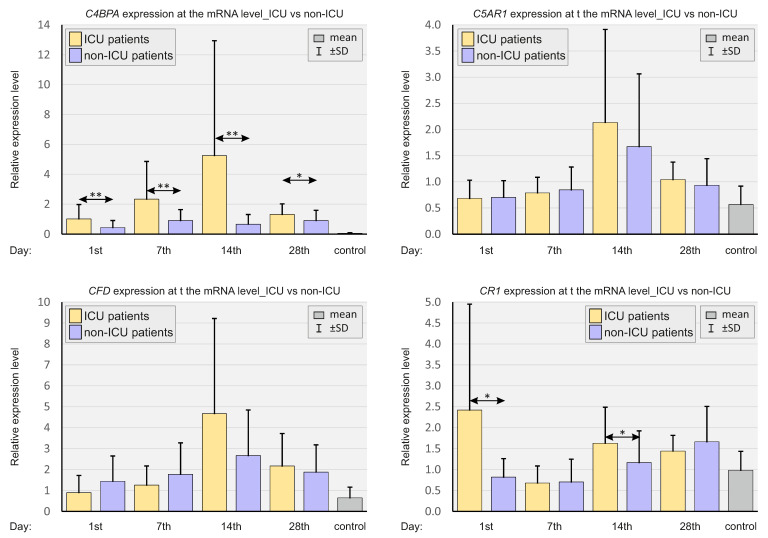
Bar graphs showing the real-time quantitation of selected genes *C4BPA*, *C5AR1*, *CFD*, and *CR1* in patients, depending on the severity of the course of COVID-19 and in SARS-CoV-2-negative controls. *p* < 0.05—*, *p* < 0.001—**.

**Figure 6 ijms-23-11338-f006:**
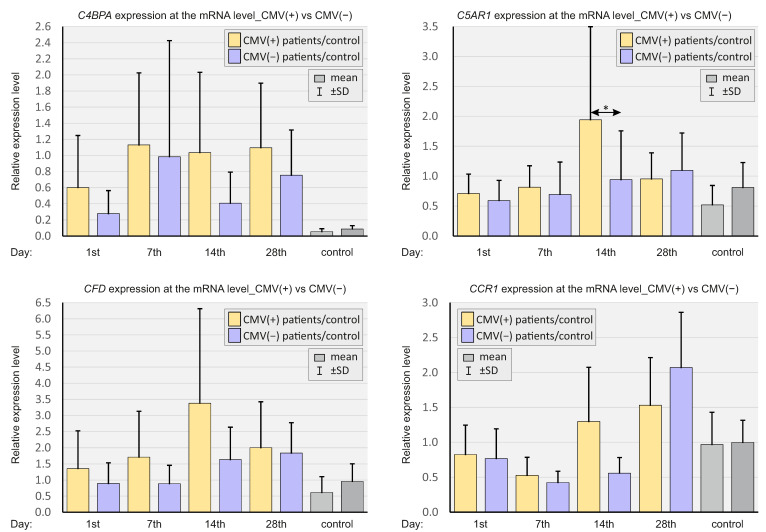
Bar graphs showing the real-time quantitation of selected genes *C4BPA*, *C5AR1*, *CFD*, and *CR1* depending on the CMV status in SARS-CoV-2-positive patients and SARS-CoV-2-negative controls. *p* < 0.05—*.

**Figure 7 ijms-23-11338-f007:**
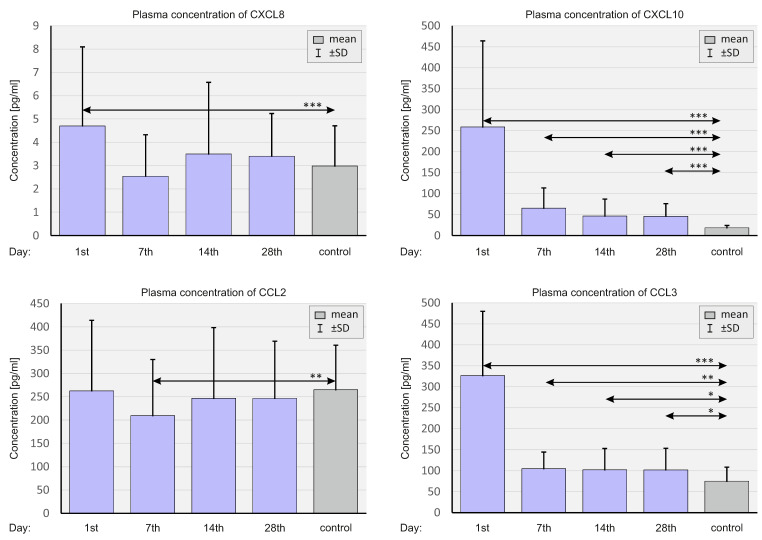
Bar graphs showing the plasma chemokine levels in the whole population of SARS-CoV-2-positive patients and SARS-CoV-2-negative controls. *p* < 0.05—*, *p* < 0.001—**, *p* < 0.0001—***.

**Figure 8 ijms-23-11338-f008:**
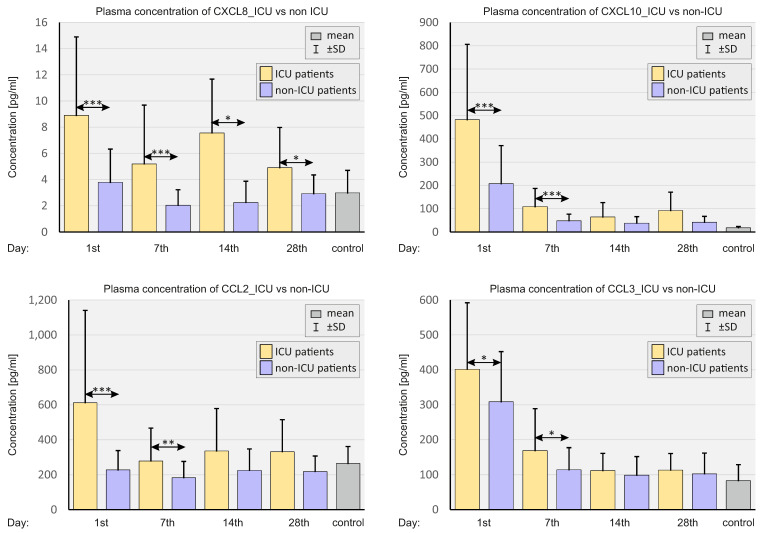
Bar graphs showing plasma chemokine levels in patients depending on the severity of the course of COVID-19 and in SARS-CoV-2-negative controls. *p* < 0.05—*, *p* < 0.001—**, *p* < 0.0001—***.

**Figure 9 ijms-23-11338-f009:**
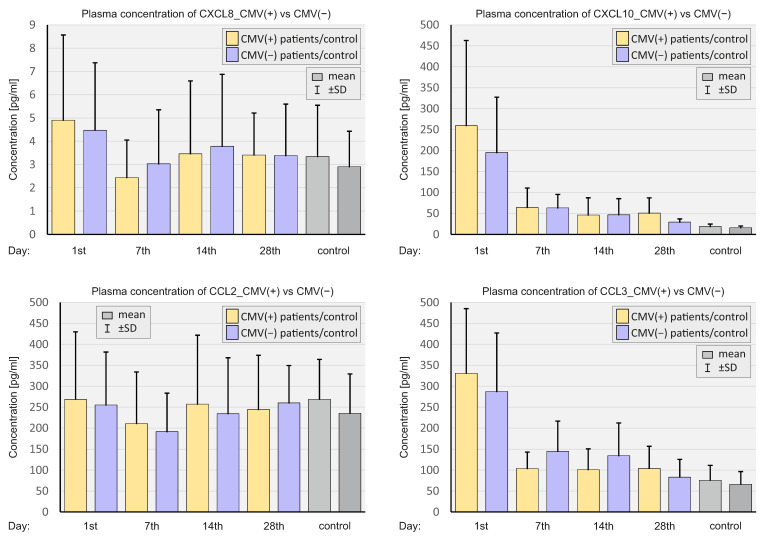
Bar graphs showing depending on the CMV status in SARS-CoV-2-positive patients and SARS-CoV-2-negative controls.

**Figure 10 ijms-23-11338-f010:**
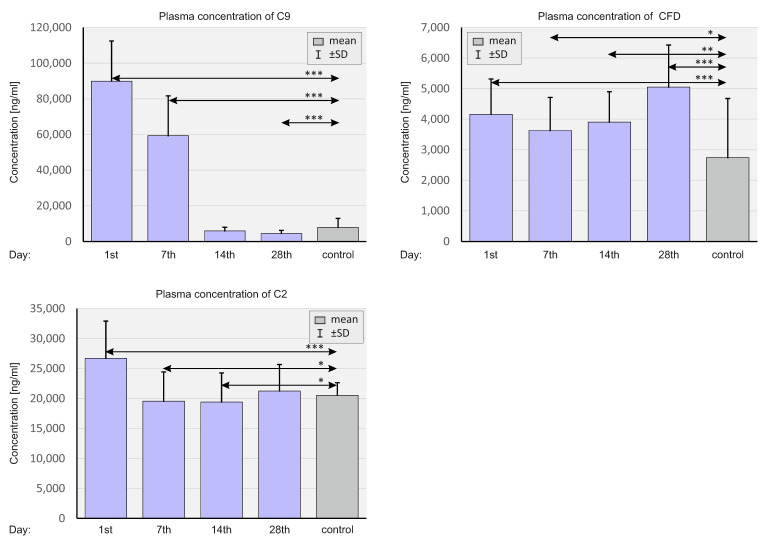
Bar graphs showing plasma complement levels in the whole population of SARS-CoV-2-positive patients and SARS-CoV-2-negative controls. *p* < 0.05—*, *p* < 0.001—**, *p* < 0.0001—***.

**Figure 11 ijms-23-11338-f011:**
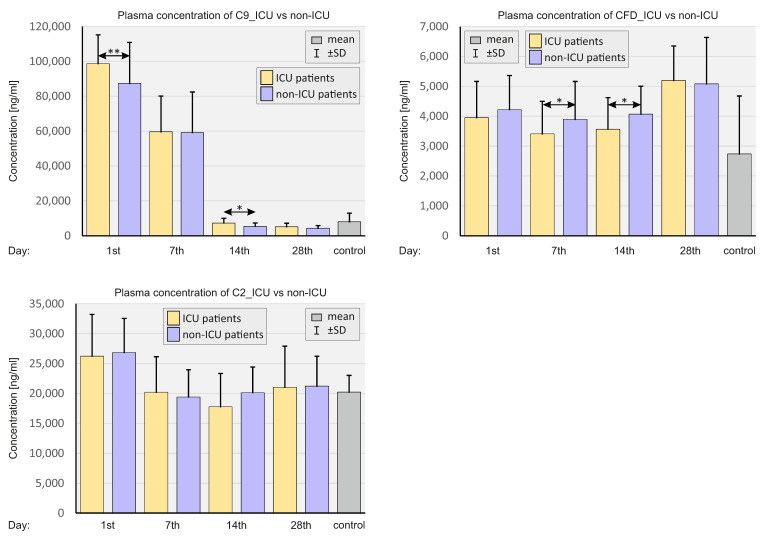
Bar graphs showing the plasma complement levels in patients depending on the severity of the course of COVID-19 and in SARS-CoV-2-negative controls. *p* < 0.05—*, *p* < 0.001—**.

**Figure 12 ijms-23-11338-f012:**
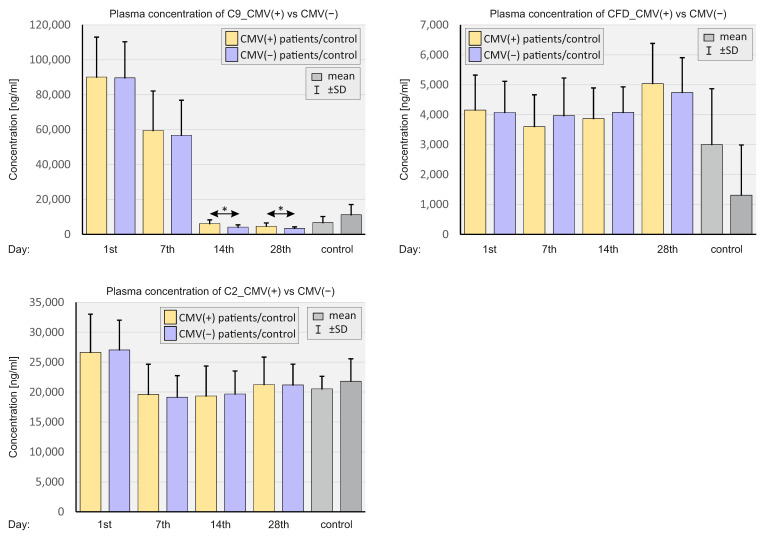
Bar graphs showing plasma complement levels depending on the CMV status in SARS-CoV-2-positive patients and SARS-CoV-2-negative controls. *p* < 0.05—*.

**Table 1 ijms-23-11338-t001:** Clinical characteristics of the study group. Statistically significant *p* values are presented in bold.

Parameter		SARS-CoV-2 Negative Controls (*n* = 80)	SARS-CoV-2 Positive Patients (*n* = 210)	*p*
Age (mean ± SD)		56.27 ± 5.56	57.78 ± 14.16	0.052
Sex (male/female)		5/75	114/96	<0.001
Body mass index (mean ± SD)		25.85 ± 4.74	29.17 ± 5.52	<0.001
Parameter		SARS-CoV-2 positive patients non-ICU (*n* = 160)	SARS-CoV-2 positive patients ICU (*n* = 50)	
Percent of patient population (%)		76.19	23.81	
Age (mean ± SD)		56.26 ± 14.53	62.64 ± 11.79	<0.01
Sex (male/female)		85/74	28/22	1
Body mass index (mean ± SD)		28.87 ± 5.55	30.44 ± 5.42	0.04
% of patients with a given parameter	Medical history			
	Hypertension	44.38	54	0.25
	Diabetes	20.63	20	1
	Ischemic heart disease	7.5	12	0.38
	Hypercholesterolemia	14.38	14	1
	Liver disease	0.63	2	0.42
	Respiratory system disease	8.13	14	0.27
	Rheumatic disease	9.38	10	1
	Cancer	10.63	10	1
	Other diseases	33.13	50	0.04
	Tobacco use (previous/now)	43.13/5.63	30/0	<0.001
	Current medications			
	Drugs taken on a permanent basis	63.75	46	0.03
	NSAIDs	12.5	16	0.49
	Statins	15	22	0.28
	Antihypertensive drugs	47.5	46	0.87
	Anticoagulants	7.5	6	1
	Cardiac drugs	7.5	14	0.17
	Anti-asthmatic drugs	8.13	10	0.77
	Other drugs	41.25	38	0.74

**Table 2 ijms-23-11338-t002:** Clinical characteristics of the control group.

Parameter		Negative Controls (n = 80)
% of patients with a given parameter	Medical history	
	Hypertension	14.08
	Diabetes	2.82
	Ischemic heart disease	0
	Hypercholesterolemia	8.45
	Liver disease	0
	Respiratory system disease	0
	Rheumatic disease	11.27
	Cancer	4.41
	Other diseases	0
	Tobacco use (previous/now)	40.26/33.80
	Current medications	
	Drugs taken on a permanent basis	21.27
	NSAIDs	8.45
	Statins	5.63
	Antihypertensive drugs	15.49
	Anticoagulants	6.63
	Cardiac drugs	0
	Anti-asthmatic drugs	2.82
	Other drugs	16.90

**Table 3 ijms-23-11338-t003:** The differences between the following time points in chemokine and complement mRNA expressions and protein concentrations in the CMV(+) patient group.

Time of Sample Collection		Day 1	Day 7	Day 14	Day 28
Groups		CMV (+) (*n* = 185)			
Number of individuals within the group		*n* = 181	*n* = 122	*n* = 69	*n* = 58
Chemokine expression		Gene relative expression level			
mean ± SD (IQR)	CXCL8	**0.32 ± 0.30 (0.21)**	**0.87 ± 0.70 (0.65) *****	**0.93 ± 0.82 (0.67) *****	**0.74 ± 0.61 (0.59) ****
	CCL2	0.55 ± 0.77 (0.23)	0.35 ± 0.38 (0.21)	0.39 ± 0.34 (0.28)	0.65 ± 0.78 (0.37)
	CCR1	**1.09 ± 0.66 (0.98)**	**0.77 ± 0.45 (0.69) ***	**0.92 ± 0,49 (0.84)**	**0.93 ± 0.48 (0.81)**
Chemokine concentration		Protein concentration (pg/mL)			
Mean ± SD (IQR)	CXCL8	**4.90 ± 3.66 (4.07)**	**2.44 ± 1.61 (2.04) ***	**3.46 ± 3.13 (1.96)**	**3.41 ± 1.8 (3.11)**
	CXCL10	**259.36 ± 203.24 (234)**	**64.11 ± 46.51 (51.76) *****	**46.06 ± 41.20 (29.12) *****	**50.59 ± 36.30 (41.82) *****
	CCL2	**268.67 ± 160.97 (232)**	**210.87 ± 123.17 (182.5) ****	**257.36 ± 164.32 (222.5)**	**244.45 ± 129.27 (221)**
	CCL3	**331.06 ± 154.05 (346)**	**103.89 ± 39.08 (92.80) *****	**101.05 ± 49.59 (92.79) *****	**104.11 ± 52.29 (80.01) *****
Complement expression		Gene relative expression level			
Mean ± SD (IQR)	C4BPA	**0.60 ± 0.65 (0.37)**	**1.13 ± 0.89 (0.99) *****	**1.04 ± 0,99 (0.62) ***	**1.10 ± 0.80 (0.84) ****
	C5AR1	**0.71 ± 0.32 (0.70)**	**0.82 ± 0.36 (0.80) ***	**1.94 ± 1.56 (1.42) *****	**0.95 ± 0.44 (0.93) ***
	CFD	**1.35 ± 1.17 (0.87)**	**1.71 ± 1.42 (1.08) ***	**3.38 ± 2.93 (2.48) *****	**2.00 ± 1.42 (1.42) ***
	CR1	**0.83 ± 0.42 (0.78)**	**0.53 ± 0.26 (0.47) ****	**1.30 ± 0.77 (1.33)**	**1.53 ± 0.68 (1.53) *****
Complementconcentration		Protein concentration (ng/mL)			
Mean ± SD (IQR)	C9	**90,070.67 ± 22,913.75 (88,300)**	**59,470.25 ± 22,633.56 (56,900) *****	**6130.59 ± 2099.12 (5730) *****	**4680.26 ± 1786.52 (4730) *****
	CFD	**4153.29 ± 1170.17 (3923.83)**	**3599.13 ± 1064.72 (3476.6) ***	**3869.71 ± 1024.44 (3817.65)**	**5038.22 ± 1344.68 (4732.5) ***
	C2	**26,629.12 ± 6388.90 (26,174.4)**	**19,602.25 ± 5044.71 (19,233) *****	**19,343.79 ± 5003.17 (19,436.8) *****	**21,253.07 ± 4590.60 (21,145.8) *****

Data are expressed as mean ± SD and median (IQR); *p* value—Friedman ANOVA for differences between day 1 and subsequent time points using Friedman ANOVA followed by Wilcoxon signed-rank tests (* *p* < 0.05, ** *p* < 0.001, *** *p* < 0.0001). For all significant differences in the Wilcoxon signed-rank test, Friedman ANOVA also yielded *p* < 0.05. Friedman ANOVA *p* values followed by significant differences in the post hoc test are shown in bold.

**Table 4 ijms-23-11338-t004:** The differences between the following time points in chemokine and complement mRNA expressions and protein concentrations in the CMV(−) patient group.

Time of sample collection		Day 1	Day 7	Day 14	Day 28
Groups		CMV (−) (n = 25)			
Number of individuals within the group		*n* = 21	*n* = 10	*n* = 10	*n* = 11
Chemokine expression		Gene relative expression level			
Mean ± SD (IQR)	CXCL8	**0.49 ± 0.40 (0.35)**	**1.77 ± 1.06 (2.22) *****	**1.67 ± 1.97 (0.76) *****	**0.85 ± 0.61 (0.83) ****
	CCL2	0.63 ± 0.83 (0.36)	0.32 ± 0.12 (0.27)	2.05 ± 2.43 (0.44)	0.85 ± 0.71 (0.58)
	CCR1	0.91 ± 0.49 (0.78)	0.82 ± 0.28 (0.68) *	1.24 ± 0.48 (1.14)	1.01 ± 0.34 (0.92)
Chemokine concentration		Protein concentration (pg/mL)			
Mean ± SD (IQR)	CXCL8	**4.47 ± 2,90 (3.93)**	**3.03 ± 2.32 (1.97) *****	**3.78 ± 3.09 (2.87)**	**3.38 ± 2.21 (2.64)**
	CXCL10	**195.20 ± 132.24 (199.5)**	**63.24 ± 31.96 (62.34) *****	**46.60 ± 38.58 (30.03) *****	**29.23 ± 7.47 (31.23) *****
	CCL2	**255.20 ± 126.41 (243)**	**191.52 ± 91.80 (202.5) ***	**234.31 ± 133.63 (200)**	**260.14 ± 89.04 (236)**
	CCL3	**287.33 ± 139.65 (284.5) *****	**144.60 ± 72.02 (119) *****	**134,54 ± 77,57 (125) *****	**83.18 ± 42.47 (80.01) *****
Complement expression		Gene relative expression level			
Mean ± SD (IQR)	C4BPA	0.28 ± 0.29 (0.18)	0.98 ± 1.44 (0.49)	0.41 ± 0.39 (0.43)	0.75 ± 0.56 (0.65)
	C5AR1	0.59 ± 0.34 (0.59)	0.69 ± 0.54 (0.74)	0.94 ± 0.82 (0.92)	1.10 ± 0.62 (1.18)
	CFD	0.89 ± 0.64 (0.82)	0.89 ± 0.57 (0.84)	1.63 ± 1.00 (1.26)	1.84 ± 0.93 (1.55)
	CR1	**0.77 ± 0.42 (0.67)**	**0.43 ± 0.16 (0.38)**	**0.56 ± 0.22 (0.55)**	**2.07 ± 0.79 (2.01) ***
Complement concentration		Protein concentration (ng/mL)			
Mean ± SD (IQR)	C9	**89,665 ± 20,673.83 (86,700)**	**56,705 ± 20,082.18 (54,875) ***	**4158.75 ± 1272.43 (4577.5) *****	**3480 ± 837.71 (3235) *****
	CFD	**4074.60 ± 1041.90 (3750.7)**	**3965.14 ± 1257.85 (3843.28) ***	**4075.83 ± 851.52 (4157.25)**	**4743.12 ± 1160.92 (4324) ***
	C2	**27,041.32 ± 4977.20 (28,002.1)**	**19,133.4 ± 3607.63 (19,404.8) *****	**19,675.76 ± 3839.96 (18381) *****	**21,176.18 ± 3482.16 (21,065.4) *****

Data are expressed as mean ± SD and median (IQR); *p* value—Friedman ANOVA for differences between day 1 and subsequent time points using Friedman ANOVA followed by Wilcoxon signed-rank tests (* *p* < 0.05, ** *p* < 0.001, *** *p* < 0.0001). For all significant differences in the Wilcoxon signed-rank test, Friedman ANOVA also yielded *p* < 0.05. Friedman ANOVA *p* values followed by significant differences in the post hoc test are shown in bold.

**Table 5 ijms-23-11338-t005:** Association of CMV status and COVID-19 severity. Fisher’s exact test, *p* value = 0.01.

Severity	CMV IgG Status	Frequency
ICU	Negative	1
Non-ICU	Negative	24
ICU	Positive	51
Non-ICU	Positive	134

## Data Availability

The data presented in this study are available on request from the corresponding author. The data are not publicly available due to ethical and privacy reasons.

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
