# Peer review of "CXCL8, CCL2, and CMV Seropositivity as New Prognostic Factors for a Severe COVID-19 Course"

_ijms, 2022, doi:10.3390/ijms231911338_

Round 1

Reviewer 1 Report

In this paper, the authors discuss the role of IL8, CCL2 and CMV seropositivity as new prognostic factors for the Covid-19 course. Although the remarkable data shown by the authors, this article requires refinement in order to be accepted.

Primarily, as mentioned by the authors as the major limitations, the sex and BMI of the patients included in this study should be better investigated and correlated with the disease status,  in order to better clarify the cytokines storm and its related immune response. Moreover, a better explanation of the huge trend differences in the mRNA expression versus protein plasma expression levels should be provided. Lastly, an additional review of grammar, sentence structure, consistency of font, and use of abbreviations is needed. 

Author Response

------------------------------------------- Reviewer #1 Comments -------------------------------------

Dear Reviewer,

Thank you for your comments concerning our manuscript entitled “IL-8, CCL2 and CMV seropositivity as new prognostic factors for severe COVID-19 course”. We have studied the comments carefully and have made corrections, which we hope, will meet with your approval. Thank you for your advice and your constructive comments.

Reviewer #1:

#1) Primarily, as mentioned by the authors as the major limitations, the sex and BMI of the patients included in this study should be better investigated and correlated with the disease status, in order to better clarify the cytokines storm and its related immune response.

(The response)

We thank the Reviewer for this comment. Gender and BMI and their impact on the course and severity of COVID-19 have been thoroughly investigated. Results of multiple studies revealed that men are more likely to suffer from severe COVID-19 than women for a variety of reasons. However, in our study we did not detect differences between men and women in terms of COVID-19 severity (p = 0.87). It is very well-established that BMI is a risk factor for severe SARS-CoV-2 infection. Results of our study are consistent with the literature. In our cohort ICU patients had significantly higher BMI that non-ICU patients (p = 0.04). To address this issue, we have modified the Results section as follows:

“The patients in the study group were significantly more often men with a higher BMI. As for patients with a more severe course of COVID-19 (ICU), they were significantly older and had higher BMI compared to non-ICU patients. We did not detect differences between men and women in terms of COVID-19 severity (p = 0.87). Interestingly, ICU patients were significantly less frequent cigarette smokers”.

#2) Moreover, a better explanation of the huge trend differences in the mRNA expression versus protein plasma expression levels should be provided.

(The response)

We wish to thank the Reviewer for the constructive comment. Indeed, the results obtained seem contradictory at first glance. Currently, the scientific literature is very limited on this topic. Most studies either focus on gene expression or on blood chemokine levels. Peterson et al. studied the patterns of gene expression in COVID-19 patients. They confirmed that the increase in biological pathways in severe COVID-19 was associated with platelet activation and coagulation, while a significant decrease was related to T-cell signaling and differentiation [Peterson DR, et al. Gene Expression Risk Scores for COVID-19 Illness Severity. J Infect Dis. 2021 Nov 30: jiab568. doi: 10.1093/infdis/jiab568]. Additionally, Arunachalam et al. showed that gene expression during COVID-19 infection is time dependent. Among others, IFN-stimulated genes show transient expression [Arunachalam PS et al. Systems biological assessment of immunity to mild versus severe COVID-19 infection in humans. Science. 2020 Sep 4;369(6508):1210-1220. doi: 10.1126/science.abc6261]. However, the only study that found a similar association to ours suggests that the discordance between IFN-α2a protein and IFNA2 transcript levels in blood suggests that type I IFNs during COVID-19 may be primarily produced by tissue-resident cells [Abers et al. An immune-based biomarker signature is associated with mortality in COVID-19 patients. JCI Insight. 2021 Jan 11;6(1):e144455. doi: 10.1172/jci.insight.144455]. In order to address this issue, we have modified the Discussion section as follows:

“These seemingly contradictory processes at the molecular and protein levels can be explained in some way. The baseline level of cytokines was assessed in patients with full-blown COVID-19 infection. The chemokines we study are produced in humans by macrophages, T lymphocytes, neutrophils, fibroblasts, osteoblasts, endothelial cells and epithelial cells. During acute infectious symptoms, the previously produced chemokines were released into the circulation. The increase in plasma chemokine concentration, through a negative feedback mechanism, most likely suppressed mRNA expression. Therefore, only a gradual decrease in the concentration of chemokines in the following days of observation resulted in a parallel increase in mRNA expression. This hypothesis seems to be supported by Abers et al., who found a similar association to ours, suggesting that the discordance between IFN-α2a protein and IFNA2 transcript levels in blood suggests that type I IFNs during COVID-19 may be primarily produced by tissue-resident cells (23). Additionally, Arunachalam et al. showed that IFN-stimulated genes show transient expression during COVID-19 infection in a time-dependent manner (24).”

#3) Lastly, an additional review of grammar, sentence structure, consistency of font, and use of abbreviations is needed.

(The response)

We wish to thank the Reviewer for the practical comment. According to the Reviewer’s request, we carefully reviewed the manuscript and all the shortcomings and errors noticed by us were corrected.

Reviewer 2 Report

The author investigated the role of CMV status, chemokines, and complement in COVID-19 at the molecular and proteomic levels. The results revealed that IL-8, CCL2, and CMV seropositivity should be considered as new prognostic factors for COVID-19 severity. The study is well conducted and the methods used are appropriate, but as they state in their limitation, the large proportion of women in the control group is a cause for concern. Could a similar trend have been shown if the experimental group had been restricted to women?

Author Response

------------------------------------------- Reviewer #2 Comments -------------------------------------

Dear Reviewer,

Thank you for your comments concerning our manuscript entitled “IL-8, CCL2 and CMV seropositivity as new prognostic factors for severe COVID-19 course”. We have studied the comments carefully and have made corrections, which we hope, will meet with your approval. Thank you for your advice and your constructive comments.

Reviewer #2:

#1) The study is well conducted and the methods used are appropriate, but as they state in their limitation, the large proportion of women in the control group is a cause for concern. Could a similar trend have been shown if the experimental group had been restricted to women?

(The response)

We wish to thank the Reviewer for the constructive comment. As requested by the Reviewer, we first checked whether there were differences between men and women in terms of the severity of COVID-19. In our study group, we did not find such a relationship (p = 0.87).

To address this issue, we have modified the Results section as follows:

“The patients in the study group were significantly more often men with a higher BMI. As for patients with a more severe course of COVID-19 (ICU), they were significantly older and had higher BMI compared to non-ICU patients. We did not detect differences between men and women in terms of COVID-19 severity (p = 0.87). Interestingly, ICU patients were significantly less frequent cigarette smokers”.

 Subsequently, we limited the study group to women and re-analyzed our data. We observed overall similar trend as in the whole group. There were only minor differences that did not affect the final conclusions. In the female cohort, we did not detect significant differences in the expression of several complement factors and in serum concentrations that were noted across the whole cohort (women and men). It should be noted, however, that such a drastic decrease in the sample size reduces the power of the statistical test, and thus increases the probability of committing a type II error. This means that the smaller the sample size, the greater the likelihood of not rejecting the null hypothesis when the alternative hypothesis is true.   

Reviewer 3 Report

Overall, this study on the seropositivity of specific cytokines and inflammatory factors in severe COVID-19 cases is potentially impactful research. The aim was to identify key factors that correlated with severe infections of SARS-CoV-2 to help detect important mechanisms involved in severe COVID-19.

Comments:

Introduction –

·        Line 59 contains the incorrect abbreviation for growth-regulated oncogene α.

·        The reason CMV status was measured needs to be clearly outlined in the introduction. Justification for looking at CMV status was lacking.

Methods –

·        The methods for COVID-19 testing should be expanded, including details on the RT-PCR for SARS-CoV-2 positivity and ELISAs.

·        It is unclear if the severe COVID-19 patients were all in the ICU for at least 14 days.

Results –

·        The results are presented in an unclear manner. All box-and-whisker plots should be changed to bar graphs to clearly visualize the significance between groups. As of now, the box-and-whisker plots make it look like there is not significance. If groups are significant, the significance and p-values should be clearly visible and labelled.

·        The entire results section needs to be expanded. Every graph presented in a figure must be discussed in the text. Summary sentences indicating what each experiment/data point suggests should be added to the end of each thought.

·        Table 1 should include the same parameters for the negative controls as the ICU and non-ICU patients, including medical history and current medications.

·        There are two different Figure 2 legends that mention different information.

·        Line 13 states there is a difference between C9 levels on day 28 while the figure does not indicate this.

·        Table 2 and 3 should not include the Friedman ANOVA p-values as these do not add information and it is confusing when looking at significance between groups.

·        The word “Expression” is misspelled in Table 2 and 3.

·        The description of the tables (the paragraphs directly below each table detailing the statistical tests and p-values) should not be included in the text. These should be included in the table legends below each table.

Discussion –

·        CCR2 is not discussed in the results or the discussion yet the expression of CCR2 was measured and represented in a figure.

CFD is mentioned in the discussion but not in the results. The data shown in the figure is not consistent with what is mentioned in the discussion. 

Author Response

------------------------------------------- Reviewer #3 Comments -------------------------------------

Dear Reviewer,

Thank you for your comments concerning our manuscript entitled “IL-8, CCL2 and CMV seropositivity as new prognostic factors for severe COVID-19 course”. We have studied the comments carefully and have made corrections, which we hope, will meet with your approval. Thank you for your advice and your constructive comments.

Reviewer #3:

#1) Introduction – Line 59 contains the incorrect abbreviation for growth-regulated oncogene α.

(The response)

We wish to thank the Reviewer for the constructive comment. According to the Reviewer’s request, we have corrected error as follows:

“Their study concluded that the cytokine profile in COVID-19 cases is peculiar to innate response (interleukin 6 (IL-6), interleukin 8 (IL-8), interleukin 1 α (IL-1 α) granulocyte colony stimulating factor (G-CSF), growth-regulated oncogene α (GROα/CXCL1), monocyte chemotactic protein-3 (MCP-3), monocyte chemoattractant protein-1 (MCP-1), tumor necrosis factor α (TNF-α).”

#2) Introduction - The reason CMV status was measured needs to be clearly outlined in the introduction. Justification for looking at CMV status was lacking.

(The response)

We wish to thank the Reviewer for the practical comment. According to the Reviewer’s request, we have added the supplementary explanation and modified the Introduction as follows:

“The role of cytokines in COVID-19 is well established. However, the exact course of the immune processes remains terra incognita. Therefore, the aim of our study was to examine the role of selected chemokines as well as the complement system in COVID-19 and how concentrations of given molecules differ over time at both mRNA and proteomic level.      

Weber and team conducted a study that investigated the relationship between disease severity and serological status of the herpes virus using statistical models. It was established that cytomegalovirus (CMV) positive non-geriatric patients were more likely to develop severe COVID-19. Their study concluded that CMV seropositivity could be a potent biomarker in identifying younger individuals at higher risk of developing severe COVID-19, especially in the absence of other comorbidities (16). Therefore, we decided to additionally investigate the status of CMV in our cohort and confront it with chemokines and complement factors.

” 

#3) Methods – The methods for COVID-19 testing should be expanded, including details on the RT-PCR for SARS-CoV-2 positivity and ELISAs.

(The response)

We wish to thank the Reviewer for the constructive comment. According to the Reviewer’s request, we have expanded the Method section as follows:

First, we removed one sentence from the section on the characteristic of the study group, which was included in the newly created section of the diagnostic of RT-PCR method.

“2.1. Study group

This retrospective cohort study included 210 patients diagnosed with COVID-19 at the Department of Infectious, Tropical Diseases and Acquired Immunodeficiency, Pomeranian Medical University in Szczecin, Poland. SARS-CoV-2 infection was confirmed from nasopharyngeal swabs with real time polymerase chain reaction (RT–PCR) technique. Tests were considered positive when at least 2 out of 3 analysed SARS-CoV-2 genes (ORF1ab, N, S) had Ct values ≤ 37.

In the next step, we expanded the Methods section with new paragraphs on ELISA and qRT-PCR as follows:

“2.3.2. RNA isolation

2.3.2.1 Viral RNA isolation

Viral RNA isolation was performed using the MagMAX Viral/Pathogen II Nucleic Acid Isolation Kit (Thermo Fisher Scientific, ON, CA) according to the manufacturer’s protocol. 200 μL of each sample was added to the designated sample well and mixed with 5 μL of proteinase K, 5 μL of MS2 phage control, 265 μL of binding buffer, and 10 μL of magnetic beads. 200 μL of nuclease-free water was also pipetted into the negative control well in the sample plate. Furthermore, 3 processing plates (KingFisher 96 Deep-Well Plate, Thermo Fisher Scientific, ON, CA) with Wash 1 Solution (500 µL per well), Wash 2 Solution—80% ethanol (1000 µL per well) and Elution Solution (50 µL per well) were also prepared. MagMAX Viral/Pathogen nucleic acid isolation was processed using an automated KingFisher Flex instrument (Thermo Fisher Scientific, ON, CA).

2.3.2.2. Blood mononuclear cell RNA isolation

Total RNA was isolated from blood mononuclear cells (MNC) using the commercial mirVana miRNA Isolation Kit (Thermo Fisher Scientific, Waltham, MA, USA). The isolation was performed according to the manufacturer’s protocol. The final concentration and quality of total RNA isolated from the cells was determined by the Epoch spectrophotometer (Biotek, Winooski, VT, USA).”

“2.3.3. qRT-PCR

2.3.3.1. qRT–PCR assays for detecting SARS-CoV-2 RNA

RT–qPCR assays for the detection of SARS-CoV-2 RNA were performed using a QuantStudio 5 PCR instrument and a TaqPath COVID 19 CE IVD RT PCR Kit (Thermo Fisher Scientific, ON, CA) according to the manufacturer’s protocol. The one-step RT–qPCR contained 5 μL of RNA template/ TaqPath COVID 19 Control, 6.25 μL of 4×TaqPath 1 Step Multiplex Master Mix (Thermo Fisher Scientific, ON, CA), 1.25 µL of COVID-19 Real Time PCR Assay Multiplex and 7.5 µL of nuclease-free water in a total volume of 20 μL. The one-step RT–qPCR program included the RT reaction at 53°C for 10 min, enzyme activation at 95°C for 2 min and 40 cycles of PCR amplification at 95°C for 3 s and 60°C for 30 s. After RT–PCR was completed, the results were analyzed using Applied Biosystems COVID-19 Interpretive Software (Thermo Fisher Scientific, ON, CA): those tested were considered positive when at least 2 out of 3 analyzed SARS-CoV-2 genes (ORF1ab, N, S) had Ct values ≤37.

2.3.3.2. qRT–PCR assays for evaluating complement and chemokine mRNA expression

Primers for selected genes (CXCL8, CCL2, C-C motif chemokine receptor 1 (CCR1), complement component 4 binding protein alpha (C4BPA), complement component 5a receptor 1 (C5AR1), complement factor D (CFD), complement receptor type 1 (CR1) were designed by BLAST PRIMER and purchased from the Laboratory of DNA Sequencing and Oligonucleotide Synthesis, Institute of Biochemistry and Biophysics, Polish Academy of Sciences, Warsaw, Poland. The following primer sequences were used: hCXCL8, f: 5’- TTCAGAGACAGCAGCAGAGCACA -3’, r: 5’- AGCACTCCTTGGCAAAACTG -3’; hCCL2, f: 5’- GATCTCAGTGCAGAGGCTCG -3’, r: 5’- TTTGCTTGTCCAGGTGGTCC -3’; hCCR1, f: 5’- AGAAGCCGGGATGGAAACTC-3’, r: 5’- TTCCAACCAGGCCAATGACA-3’; hC4BPA, f: 5’- AGGGACTCTTTGGTGGAGCA -3’, r: 5’- CTGCTGCTTCGCTGATGTTT -3’; hC5AR1, f: 5’- AGCCCAGGAGACCAGAACAT -3’, r: 5’- CACCAGGAAGACGACTGCAA -3’; hCFD, f: 5’- GATGTGCGCGGAGAGCAAT -3’, r: 5’- CTGTCGATCCAGGCCGCATA -3’; hCR1, f: 5’- TCTGCTGTCTTGGGTGCATT -3’, r: 5’- TTCGTGATGATTCTGCCCCC -3’; hBMG, f: 5’- AATGCGGCATCTTCAAACCT -3’, r: 5’- TGACTTTGTCACAGCCCAAGA-3’. Gene expression studies were performed on the Bio-Rad CFX96 Real-Time PCR Detection System (Bio-Rad Inc., Hercules, CA, USA).”

“2.3.5. Enzyme linked immunosorbent assay (ELISA) - serological assays for specific anti-SARS-CoV-2 IgM, IgG, IgA and anti-CMV IgG antibodies detection

All the plasma samples were tested for the presence of anti-SARS-CoV-2 immunoglobulins (IgM, IgG, IgA) and also for anti-cytomegalovirus (anti-CMV) IgG antibodies using a commercially available enzyme-linked immunosorbent assay according to the manufacturer’s instructions (EUROIMMUN AG, Lubeck, Germany). First, the reagent wells on the microplate strips that had previously been coated with inactivated CMV antigen and recombinant structural spike protein of SARS-CoV-2 were filled with the diluted patient plasma sample and allowed to incubate. Specific anti-CMV IgG and anti-SARS-CoV-2 IgM, IgG and IgA antibodies, attached to the coated wells, were identified with peroxidase-labeled anti-human IgM/IgG/IgA antibodies. The staining intensity generated after hydrolysis of the peroxidase substrate was measured using Varioskan LUX multimode microplate reader (TSA, Thermo Fisher Scientific, Waltham, MA, USA) at a wavelength of 450 nm. The results were interpreted according to the manufacturer’s instructions: samples with a concentration higher than 22 RU/ml were considered positive for CMV IgG, while samples below the cut-off value were considered negative. The results for SARS-CoV-2 specific antibodies were evaluated semi-quantitatively by calculation of the ratio (R) of the extinction of the control or patient sample over the extinction of the calibrator. Following the manufacturer’s instruction, the level of positivity was calculated as the R between the absorbance values of samples and calibrator, at a wavelength of 450nm (R < 0.8, negative; 0.8 < R < 1.1, weakly positive; R ≥ 1.1, strongly positive).

#4) Methods - It is unclear if the severe COVID-19 patients were all in the ICU for at least 14 days.

(The response)

We wish to thank the Reviewer for the constructive comment. The criteria for including patients in the ICU group were at least one of the following:

1) required ICU admission due to respiratory failure

2) the disease was fatal

3) required hospitalization longer than 14 days directly related to COVID-19 (this group excluded patients whose extended hospitalization time was evidently due to other reasons, e.g. severe pancreatitis or epidemiological reasons; in the first months of the pandemic, patients could be dismissed from isolation only after obtaining two negative RT-PCR results for SARS-CoV-2, which contributed to the extension of hospitalization time and was not necessarily associated with the severe course of the disease). Therefore, if at least one of above criteria was met, the patient was included in the ICU group. Thus, each of these criteria, individually or in combination, determined the inclusion of patients in the ICU group.

To be more precise, we have modified the Materials and Methods section as follows:

“Group 1 (intensive care unit patients - ICU patients), included patients who required ICU admission due to respiratory failure, or hospitalization longer than 14 days directly related to COVID-19, or the disease was fatal. Meeting each of these criteria, individually or in combination, determined the inclusion of patients in the ICU group. Group 2 (non-intensive care unit patients – non-ICU patients), included asymptomatic or mildly symptomatic patients who had oxygen saturation of at least 95% and did not require hospitalization due to COVID-19 and symptomatic patients with oxygen saturation lower than 95% who required hospital admission up to fourteen days.”

#5) Results – The results are presented in an unclear manner. All box-and-whisker plots should be changed to bar graphs to clearly visualize the significance between groups. As of now, the box-and-whisker plots make it look like there is not significance. If groups are significant, the significance and p-values should be clearly visible and labelled.

(The response)

We thank the Reviewer for this comment.

Figure 1. Bar graphs showing real-time quantitation of selected genes CXCL8, CCL2, CCR1 in whole population of SARS-CoV-2-positive patients and SARS-CoV2-negative controls. p<0.05 - *, p<0.001 - **, p<0.0001 - ***.

Figure 2. Bar graphs showing real-time quantitation of selected genes CXCL8, CCL2, CCR1 in patients depending on the severity of the course of COVID-19 and in SARS-CoV2-negative controls. p<0.05 - *, p<0.001 - **, p<0.0001 - ***.

Figure 3. Bar graphs showing real-time quantitation of selected genes CXCL8, CCL2, CCR1 depending on the CMV status in SARS-CoV2-positive patients and SARS-CoV-2-negative controls. p<0.05 - *, p<0.001 - **, p<0.0001 - ***.

Figure 4. Bar graphs showing real-time quantitation of selected genes C4BPA, C5AR1, CFD, CR1 in whole population of SARS-CoV-2-positive patients and SARS-CoV2-negative controls. p<0.05 - *, p<0.001 - **, p<0.0001 - ***.

Figure 5. Bar graphs showing real-time quantitation of selected genes C4BPA, C5AR1, CFD, CR1 in patients depending on the severity of the course of COVID-19 and in SARS-CoV2-negative controls. p<0.05 - *, p<0.001 - **, p<0.0001 - ***.

Figure 6. Bar graphs showing real-time quantitation of selected genes C4BPA, C5AR1, CFD, CR1 depending on the CMV status in SARS-CoV2-positive patients and SARS-CoV-2-negative controls. p<0.05 - *, p<0.001 - **, p<0.0001 - ***.

Figure 7. Bar graphs showing plasma chemokine levels in whole population of SARS-CoV-2-positive patients and SARS-CoV2-negative controls. p<0.05 - *, p<0.001 - **, p<0.0001 - ***.

Figure 8. Bar graphs showing plasma chemokine levels in patients depending on the severity of the course of COVID-19 and in SARS-CoV2-negative controls. p<0.05 - *, p<0.001 - **, p<0.0001 - ***.

Figure 9. Bar graphs showing plasma chemokine levels depending on the CMV status in SARS-CoV2-positive patients and SARS-CoV-2-negative controls. p<0.05 - *, p<0.001 - **, p<0.0001 - ***.

Figure 10. Bar graphs showing plasma complement levels in whole population of SARS-CoV-2-positive patients and SARS-CoV2-negative controls. p<0.05 - *, p<0.001 - **, p<0.0001 - ***.

Figure 11. Bar graphs showing plasma complement levels in patients depending on the severity of the course of COVID-19 and in SARS-CoV2-negative controls. p<0.05 - *, p<0.001 - **, p<0.0001 - ***.

Figure 12. Bar graphs showing plasma complement levels depending on the CMV status in SARS-CoV2-positive patients and SARS-CoV-2-negative controls. p<0.05 - *, p<0.001 - **, p<0.0001 - ***.

#6) Results - The entire results section needs to be expanded. Every graph presented in a figure must be discussed in the text. Summary sentences indicating what each experiment/data point suggests should be added to the end of each thought.

(The response)

We wish to thank the Reviewer for the constructive comment which helps to improve the paper. According to the generally accepted principle, we describe the obtained results only statistically significant. Similarly, we formulate conclusions from the results obtained in the Discussion section.

We have modified the Results section as follows:

“The relative expression level of CXCL8 and CCL2 was significantly higher in the control group than in the study group at all time points. However, from day 7 onwards, a virtually constant upward trend in CXCL8 and CCL2 expression is apparent. Relative CCR1expression was significantly lower in the study group compared to the control.”

“CXCL8 expression was significantly lower on days 1, 7, 14, and concurrent CCL2 expression was significantly lower on days 1 and 7 in patients with more severe course of COVID-19. While CCR1 expression was significantly lower only on day 7 in ICU patients”.

“CXCL8 expression was significantly lower on day 7 in CMV SARS-CoV2-positive patients compared to CMV negative patients. There is visible an overall trend of higher expression of CXCL8 and CCL2 in both patient and control groups with negative CMV status”.

“The relative expression of C4BPA, C5AR1 and CFD was substantially higher in the study group at all timepoints compared to control. There is a clear upward trend in the expression of the examined complement elements from day 1 to 14. CR1 expression was significantly lower on day 7 and higher on day 28 in the study group compared to the control group”.

“C4BPA expression was higher at all time points in the ICU patients compared to non-ICU patients. While CR1 expression was substantially higher in the ICU patients on day 1 and 14 compared to non-ICU patients.”

“In general, concentration of CXCL10 and CCL3 was significantly higher at all time points in the study group compared to the control group. At the same time, a uniform downward trend in the concentration of these chemokines in the study group was also demonstrated at subsequent time points. The initial plasma concentration of CXCL8 on day 1 and CCL2 on day 7 were substantially higher in SARS-CoV-2-positive patients than SARS-CoV2-negative controls.”

“The mean concentrations of all assessed chemokines were higher on day 1 in CMV(+) SARS-CoV-2-positive patients compared to CMV(-) SARS-CoV-2-positive and SARS-CoV-2-negative patients, however statistically insignificant.”

“The baseline plasma concentrations of the tested complement elements were significantly higher in population of SARS-CoV-2-positive patients compared to control group. There is a downward trend in the concentration of complement elements to the 14th day. The plasma concentration of C9 on days 1, 7, CFD at all time points, and C2 on day 1 were significantly higher in the study group compared to the control group. While concentrations of C9 on day 28, and C2 on day 14 were significantly lower in SARS-CoV-2-positive patients compared to SARS-CoV-2-negative patients.”

“Plasma concentration of C9 was higher in ICU patients on days 1 and 14 compared to non-ICU patients. Concentration of CFD was substantially lower in ICU patients on days 7 and 14 compared to non-ICU patients.”

“We observed very high concentration of C9 on days 1 and 7 in CMV(+) and CMV(-) SARS-CoV-2-positive group compared to control. However, the differences between CMV(+) and CMV(-) SARS-CoV-2-positive patients were statistically insignificant. On days 14 and 28, a sharp decrease in C9 concentration can be observed with significant differences between CMV(+) and CMV(-) groups”. 

We have modified also Discussion section as follows:

“On the other hand, the plasma concentration of CXCL8 on day 1 and CCL2 on day 7 were significantly higher in SARS-CoV-2 positive group on day 1 and 7, respectively, than in the control group. Lower baseline CXCL8 and CCL2 expression correlated with a more severe course of COVID. While, increased CXCL8 plasma concentration on all days and CCL2 on days 1 and 7 also correlated with a more severe course of COVID.”

#7) Results - Table 1 should include the same parameters for the negative controls as the ICU and non-ICU patients, including medical history and current medications.

(The response)

We thank the Reviewer for this comment. We added the following table including descriptive statistics of medical history and current medications for the control group:

“The patients in the study group were significantly more often men with a higher BMI. As for patients with a more severe course of COVID-19 (ICU), they were significantly older and had higher BMI compared to non-ICU patients. Interestingly, ICU patients were significantly less frequent cigarette smokers. Table 2 presents clinical characteristics of the control group.

Table 2. Clinical characteristics of the control group.

Parameter

Negative controls

(n=80)

% of patients with a given parameter

Medical history:

Hypertension

14.08

Diabetes

2.82

Ischemic heart disease

0

Hypercholesterolemia

8.45

Liver disease

0

Respiratory system disease

0

Rheumatic disease

11.27

Cancer

4.41

Other diseases

0

Tobacco use (previous/now)

40.26/33.80

Currently taken medications:

Drugs taken on permanent basis

21.27

NSAIDs

8.45

Statins

5.63

Antihypertensive drugs

15.49

Anticoagulants

6.63

Cardiac drugs

0

Anti-asthmatic drugs

2.82

Other drugs

16.90

#8) Results - There are two different Figure 2 legends that mention different information.

(The response)

We wish to thank the Reviewer for the practical comment. We have modified the Results section as follows:

Figure 2. shows a comparison of the relative expression of chemokines depending on the severity of the course of COVID-19.

Figure 2. Bar graphs showing real-time quantitation of selected genes CXCL8, CCL2, CCR1 in patients depending on the severity of the course of COVID-19 and in SARS-CoV2-negative controls. p<0.05 - *, p<0.001 - **, p<0.0001 - ***.

#9) Results - Line 13 states there is a difference between C9 levels on day 28 while the figure does not indicate this.

(The response)

We wish to thank the Reviewer for the constructive comment. We have corrected the description in the Results section as follows:

“Plasma concentration of C9 was higher in ICU patients on days 1 and 14 compared to non-ICU patients”.

#10) Results - Table 2 and 3 should not include the Friedman ANOVA p-values as these do not add information and it is confusing when looking at significance between groups.

(The response)

We wish to thank the Reviewer for the constructive comment which helps to improve the paper. According to the Reviewer’s request, we have modified Table 3 and 4 and deleted the Friedman ANOVA p-values. Comparisons in which post-hoc analysis yielded significant results are shown in bold. Here we present the modified Tables:

Table 3. The differences between the following time points in chemokine and complement mRNA expression and protein concentration in the CMV(+) group of patients.

Time of sample collection

Day 1

Day 7

Day 14

Day 28

Groups

CMV (+) (n = 185)

Number of individuals within group

n = 181

n = 122

n = 69

n = 58

Chemokine

expression

Gene relative expression level

mean ± SD (IQR)

CXCL8

0.32 ± 0.30 (0.21)

0.87 ± 0.70 (0.65) ***

0.93 ± 0.82 (0.67) ***

0.74 ± 0.61 (0.59) **

CCL2

0.55 ± 0.77 (0.23)

0.35 ± 0.38 (0.21)

0.39 ± 0.34 (0.28)

0.65 ± 0.78 (0.37)

CCR1

1.09 ± 0.66 (0.98)

0.77 ± 0.45 (0.69) *

0.92 ± 0,49 (0.84)

0.93 ± 0.48 (0.81)

Chemokine

concentration

Protein concentration (pg/mL)

mean ± SD (IQR)

CXCL8

4.90 ± 3.66 (4.07)

2.44 ± 1.61 (2.04)*

3.46 ± 3.13 (1.96)

3.41 ± 1.8 (3.11)

CXCL10

259.36 ± 203.24 (234)

64.11 ± 46.51 (51.76) ***

46.06 ± 41.20
(29.12) ***

50.59 ± 36.30 (41.82) ***

CCL2

268.67 ± 160.97 (232)

210.87 ± 123.17 (182.5) **

257.36 ± 164.32 (222.5)

244.45 ± 129.27 (221)

CCL3

331.06 ± 154.05 (346)

103.89 ± 39.08
(92.80) ***

101.05 ± 49.59
(92.79) ***

104.11 ± 52.29
(80.01) ***

Complement

expression

Gene relative expression level

mean ± SD (IQR)

C4BPA

0.60 ± 0.65 (0.37)

1.13 ± 0.89 (0.99) ***

1.04 ± 0,99 (0.62) *

1.10 ± 0.80 (0.84) **

C5AR1

0.71 ± 0.32 (0.70)

0.82 ± 0.36
(0.80) *

1.94 ± 1.56
(1.42) ***

0.95 ± 0.44
(0.93) *

CFD

1.35 ± 1.17 (0.87)

1.71 ± 1.42 (1.08) *

3.38 ± 2.93 (2.48) ***

2.00 ± 1.42 (1.42) *

CR1

0.83 ± 0.42 (0.78)

0.53 ± 0.26 (0.47) **

1.30 ± 0.77 (1.33)

1.53 ± 0.68 (1.53) ***

Complement

concentration

Protein concentration (ng/mL)

mean ± SD (IQR)

C9

90070.67 ± 22913.75 (88300)

59470.25 ± 22633.56 (56900) ***

6130.59 ± 2099.12 (5730) ***

4680.26 ± 1786.52 (4730) ***

CFD

4153.29 ± 1170.17 (3923.83)

3599.13 ± 1064.72 (3476.6) *

3869.71 ± 1024.44 (3817.65)

5038.22 ± 1344.68 (4732.5) *

C2

26629.12 ± 6388.90 (26174.4)

19602.25 ± 5044.71 (19233) ***

19343.79 ± 5003.17 (19436.8) ***

21253.07 ± 4590.60 (21145.8) ***

Data are expressed as mean ± SD and median (IQR); p value — Friedman ANOVA for difference between day 1 and subsequent time points using Friedman ANOVA followed by Wilcoxon signed-rank tests (* p < 0.05, ** p < 0.001, *** p < 0.0001). For all differences significant in the Wilcoxon signed-rank test, Friedman ANOVA also yielded p < 0.05. Friedman ANOVA p values followed by significant differences in the post hoc test are shown in bold.

Table 4. The differences between the following time points in chemokine and complement mRNA expression and protein concentration in the CMV(-) group of patients.

Time of sample collection

Day 1

Day 7

Day 14

Day 28

Groups

CMV (-) (n = 25)

Number of individuals within group

n = 21

n = 10

n = 10

n = 11

Chemokine

Expression

Gene relative expression level

mean ± SD (IQR)

CXCL8

0.49 ± 0.40 (0.35)

1.77 ± 1.06 (2.22) ***

1.67 ± 1.97 (0.76) ***

0.85 ± 0.61 (0.83) **

CCL2

0.63 ± 0.83 (0.36)

0.32 ± 0.12 (0.27)

2.05 ± 2.43 (0.44)

0.85 ± 0.71 (0.58)

CCR1

0.91 ± 0.49 (0.78)

0.82 ± 0.28 (0.68) *

1.24 ± 0.48 (1.14)

1.01 ± 0.34 (0.92)

Chemokine concentration

Protein concentration (pg/mL)

mean ± SD (IQR)

CXCL8

4.47 ± 2,90 (3.93)

3.03 ± 2.32 (1.97) ***

3.78 ± 3.09 (2.87)

3.38 ± 2.21 (2.64)

CXCL10

195.20 ± 132.24 (199.5)

63.24 ± 31.96 (62.34) ***

46.60 ± 38.58 (30.03) ***

29.23 ± 7.47 (31.23) ***

CCL2

255.20 ± 126.41 (243)

191.52 ± 91.80 (202.5) *

234.31 ± 133.63 (200)

260.14 ± 89.04 (236)

CCL3

287.33 ± 139.65 (284.5) ***

144.60 ± 72.02 (119) ***

134,54 ± 77,57 (125) ***

83.18 ± 42.47 (80.01) ***

Complement expression

Gene relative expression level

mean ± SD (IQR)

C4BPA

0.28 ± 0.29 (0.18)

0.98 ± 1.44 (0.49)

 0.41 ± 0.39 (0.43)

0.75 ± 0.56 (0.65)

C5AR1

0.59 ± 0.34 (0.59)

0.69 ± 0.54 (0.74)

0.94 ± 0.82 (0.92)

1.10 ± 0.62 (1.18)

CFD

0.89 ± 0.64 (0.82)

0.89 ± 0.57 (0.84)

1.63 ± 1.00 (1.26)

1.84 ± 0.93 (1.55)

CR1

0.77 ± 0.42 (0.67)

0.43 ± 0.16 (0.38)

0.56 ± 0.22 (0.55)

2.07 ± 0.79 (2.01) *

Complement concentration

Protein concentration (ng/mL)

mean ± SD (IQR)

C9

89665 ± 20673.83 (86700)

56705 ± 20082.18 (54875) *

4158.75 ± 1272.43 (4577.5) ***

3480 ± 837.71 (3235) ***

CFD

4074.60 ± 1041.90 (3750.7)

3965.14 ± 1257.85 (3843.28) *

4075.83 ± 851.52 (4157.25)

4743.12 ± 1160.92 (4324) *

C2

27041.32 ± 4977.20 (28002.1)

19133.4 ± 3607.63 (19404.8) ***

19675.76 ± 3839.96 (18381) ***

21176.18 ± 3482.16 (21065.4) ***

Data are expressed as mean ± SD and median (IQR); p value — Friedman ANOVA for difference between day 1 and subsequent time points using Friedman ANOVA followed by Wilcoxon signed-rank tests (* p < 0.05, ** p < 0.001, *** p < 0.0001). For all differences significant in the Wilcoxon signed-rank test, Friedman ANOVA also yielded p < 0.05. Friedman ANOVA p values followed by significant differences in the post hoc test are shown in bold.

#11) Results - The word “Expression” is misspelled in Table 2 and 3.

(The response)

We thank the Reviewer for this comment. This error was corrected.

#12) Results - The description of the tables (the paragraphs directly below each table detailing the statistical tests and p-values) should not be included in the text. These should be included in the table legends below each table.

(The response)

We wish to thank the Reviewer for the practical comment. According to the Reviewer’s suggestion, we have distinguished a dedicated legend under each of the figures. Information on statistical significance between the studied groups was included only in the legend below the figure. Originally, the legend was placed imprecisely within the figure area. Modified figures and legends were presented in one of the previous comments to the Reviewer.

#13) Discussion – CCR1 is not discussed in the results or the discussion yet the expression of CCR1 was measured and represented in a figure.

(The response)

We thank the Reviewer for this comment. We have modified the Results section as follows:

“The relative expression level of CXCL8 and CCL2 was significantly higher in the control group than in the study group at all time points. However, from day 7 onwards, a virtually constant upward trend in CXCL8 and CCL2 expression is apparent. Relative CCR1 expression was significantly lower in the study group compared to the control.”

“CXCL8 expression was significantly lower on days 1, 7, 14, and concurrent CCL2 expression was significantly lower on days 1 and 7 in patients with more severe course of COVID-19. While CCR1 expression was significantly lower only on day 7 in ICU patients”.

We also modified the Discussion section adding information regarding CCR1:

“However, SARS-CoV-2 positive individuals had higher plasma concentrations of CXCL10, CCL5 and GM-CSF, which was also associated with a longer duration of mechanical ventilation (25).

Stikker et al. investigated the 3p21.31 locus and subsequent downstream signaling with regards to the COVID-19 severity. They demonstrated that several 3p21.31 variants, previously identified by applying genome-wide association studies, are associated with increased CCR1, CCR2, CCR3 and CCR5 expression. CCR1, CCR2 and CCR5 upregulation could enhance lung infiltration by monocytes and macrophages upon viral infection and mediate hyperinflammation and organ damage in the aftermath (32). However, we did not observe such association.

Contrary to IL-8, IL-10 exerts anti-inflammatory properties. Therefore, it plays a key role in infection, limiting the immune response to pathogens and thus preventing damage to the host (26).”   

#14) Discussion - CFD is mentioned in the discussion but not in the results. The data shown in the figure is not consistent with what is mentioned in the discussion. 

(The response)

We thank the Reviewer for this comment. We have modified the Results section as follows:

“The baseline plasma concentrations of the tested complement elements were significantly higher in population of SARS-CoV-2-positive patients compared to control group. There is a downward trend in the concentration of complement elements to the 14th day. The plasma concentration of C9 on days 1, 7, CFD at all time points, and C2 on days 1, 7 were significantly higher in the study group compared to the control group. While concentrations of C9 on day 28, and C2 on day 14 were significantly lower in SARS-CoV-2-positive patients compared to SARS-CoV-2-negative patients.”

“Plasma concentration of C9 was higher in ICU patients on days 1 and 14 compared to non-ICU patients. Concentration of CFD was substantially lower in ICU patients on days 7 and 14 compared to non-ICU patients.”

Now the CDF results are in line with the discussion.

Reviewer 4 Report

The manuscript by Ewa et al. reports the results of a study based on retrospective cohort design. The study design is clear, results are well described, conclusions are draw based on the results presented. Overall, the manuscript is well written and organized. However, I have two minor comments. 

- In the abstract, please write the examined chemokines and complements. 

- The introduction must include an adequate background about the examined chemokines and complements and their roles in COVID-19. Also, the authors should state why they selected these markers?

- How did the authors calculate sample size? and what about the matching?

- I cannot find primer sequences.

Author Response

------------------------------------------- Reviewer #4 Comments -------------------------------------

Dear Reviewer,

Thank you for your comments concerning our manuscript entitled “IL-8, CCL2 and CMV seropositivity as new prognostic factors for severe COVID-19 course”. We have studied the comments carefully and have made corrections, which we hope, will meet with your approval. Thank you for your advice and your constructive comments.

Reviewer #4:

#1) In the abstract, please write the examined chemokines and complements. 

(The response)

We wish to thank the Reviewer for the practical comment. According to the Reviewer’s suggestion, we have modified the Abstract section as follows:

“Concentrations of selected chemokines (CXCL8, CXCL10, CCL2, CCL3, CCR1) and complement factors (C2, C9, CFD, C4BPA, C5AR1, CR1) were examined at mRNA and protein level with regards to COVID-19 course (ICU vs non-ICU group) and CMV status at different time intervals”.

#2) The introduction must include an adequate background about the examined chemokines and complements and their roles in COVID-19. Also, the authors should state why they selected these markers?

(The response)

We wish to thank the Reviewer for the useful comment. As requested by the Reviewer, we have provided an additional explanation regarding the arguments for selecting adequate biomarkers in COVID-19. We modified the Introduction section as follows:

“What is more, severe COVID‐19 patients who received high‐flow oxygen inhalation and mechanical ventilation during hospitalization had significantly higher baseline IL‐6 levels than those without the need for oxygen therapy support (10).   

Abers et al. confirmed that CCL2 and interleukin-10 (IL-10) were associated with increased mortality in patients with COVID-19 (11). Another study by Sacks et al. concluded that highest expression levels (measured at mRNA level) of dipeptidyl peptidase 9 (DPP9) and CCR2 were observed in severe forms of COVID-19 disease (12). Moreover, similar observation applied to C-C motif Chemokine Ligand 3 (CCL3/MIP-1α) as well (13). Given the above research, we decided to focus on the following biomarkers – CXCL8, CCL2 and C-C motif chemokine receptor 1 (CCR1) expression, and serum concentrations of CXCL8, C-X-C motif chemokine ligand 10 (CXCL10/IP-10), CCL2 and CCL3, as they appear to contribute to the pathogenesis of severe COVID-19.”

#3) How did the authors calculate sample size? and what about the matching?

(The response)

We wish to thank the Reviewer for the practical comment. Our study was conducted during the COVID-19 pandemic wave, which made its design difficult. For our study, we recruited patients who met predetermined criteria, were ready to participate, and were able to sign informed consent. In the end, 210 individuals were enrolled, but this number has not been determined earlier. Due to above-mentioned reasons, a proper matching was also hindered. We enrolled healthy health care workers in the control group because during the pandemic and the regulations and restrictions related to COVID-19, they were at that time the only possible population for recruitment.   

#4) I cannot find primer sequences.

(The response)

Thank you for this suggestion. We added primer sequences in the Methods section that we already mentioned in the previous answer for Reviewer 2, as follows:

“The following primer sequences were used: hCXCL8, f: 5’- TTCAGAGACAGCAGCAGAGCACA -3’, r: 5’- AGCACTCCTTGGCAAAACTG -3’; hCCL2, f: 5’- GATCTCAGTGCAGAGGCTCG -3’, r: 5’- TTTGCTTGTCCAGGTGGTCC -3’; hCCR1, f: 5’- AGAAGCCGGGATGGAAACTC-3’, r: 5’- TTCCAACCAGGCCAATGACA-3’; hC4BPA, f: 5’- AGGGACTCTTTGGTGGAGCA -3’, r: 5’- CTGCTGCTTCGCTGATGTTT -3’; hC5AR1, f: 5’- AGCCCAGGAGACCAGAACAT -3’, r: 5’- CACCAGGAAGACGACTGCAA -3’; hCFD, f: 5’- GATGTGCGCGGAGAGCAAT -3’, r: 5’- CTGTCGATCCAGGCCGCATA -3’; hCR1, f: 5’- TCTGCTGTCTTGGGTGCATT -3’, r: 5’- TTCGTGATGATTCTGCCCCC -3’; hBMG, f: 5’- AATGCGGCATCTTCAAACCT -3’, r: 5’- TGACTTTGTCACAGCCCAAGA-3’.”

We would like to thank the Reviewers for the helpful comments and hope that our manuscript is now improved and better represents our work. We hope that the revised manuscript is acceptable for publication in International Journal of Molecular Sciences.
